# $\Delta$Energy: Optimizing Energy Change During Vision-Language Alignment Improves both OOD Detection and OOD Generalization

**Lin Zhu[1], Yifeng Yang[1], Xinbing Wang[1], Qinying Gu[2], Nanyang Ye[1]** [*]

[1] Shanghai Jiao Tong University, [2] Shanghai Artificial Intelligence Laboratory

{zhulin_sjtu, xwang8, ynylincoln}@sjtu.edu.cn,
maxwellquadyang@gmail.com
guqinying@pjlab.org.cn

## Abstract

Recent approaches for vision-language models (VLMs) have shown remarkable success in achieving fast downstream adaptation. When applied to real-world downstream tasks, VLMs inevitably encounter both the in-distribution (ID) data and out-of-distribution (OOD) data. The OOD datasets often include both covariate shifts (e.g., known classes with changes in image styles) and semantic shifts (e.g., test-time unseen classes). This highlights the importance of improving VLMs' generalization ability to covariate-shifted OOD data, while effectively detecting open-set semantic-shifted OOD classes. In this paper, inspired by the substantial energy change observed in closed-set data when re-aligning vision-language modalities—specifically by directly reducing the maximum cosine similarity to a low value—we introduce a novel OOD score, named $\Delta$Energy. $\Delta$Energy significantly outperforms the vanilla energy-based OOD score and provides a more reliable approach for OOD detection. Furthermore, $\Delta$Energy can simultaneously improve OOD generalization under covariate shifts, which is achieved by lower-bound maximization for $\Delta$Energy (termed EBM). EBM is theoretically proven to not only enhance OOD detection but also yields a domain-consistent Hessian, which serves as a strong indicator for OOD generalization. Based on this finding, we developed a unified fine-tuning framework that allows for improving VLMs' robustness in both OOD generalization and OOD detection. Extensive experiments on challenging OOD detection and generalization benchmarks demonstrate the superiority of our method, outperforming recent approaches by **10%–25%** in AUROC.

## 1 Introduction

Recent advances in pre-trained vision-language models (VLMs), such as CLIP (Radford et al., 2021), VLMo (Bao et al., 2022), MiniGPT-4 (Zhu et al., 2023a), etc., have shown promising results in visual-semantic learning. However, downstream use cases often involve further fine-tuning of VLMs. When applied to real-world downstream tasks, VLMs inevitably face challenges related to out-of-distribution (OOD) data, stemming from differences in data distributions between the training and test sets (Meinshausen and Bühlmann, 2015; Koh et al., 2020). As illustrated in Figure 1, these OOD datasets often involve *closed-set OOD* data that exhibit *covariate shifts* (i.e., changes in environments, while class labels remain the same as the in-distribution data), as well as *open-set OOD* data with *semantic shifts* (i.e., test-time new categories that were unseen during fine-tuning). It is crucial to distinguish these unknown categories from known ones, rather than blindly predicting them as known

---

[*]Nanyang Ye is the corresponding author.

39th Conference on Neural Information Processing Systems (NeurIPS 2025).

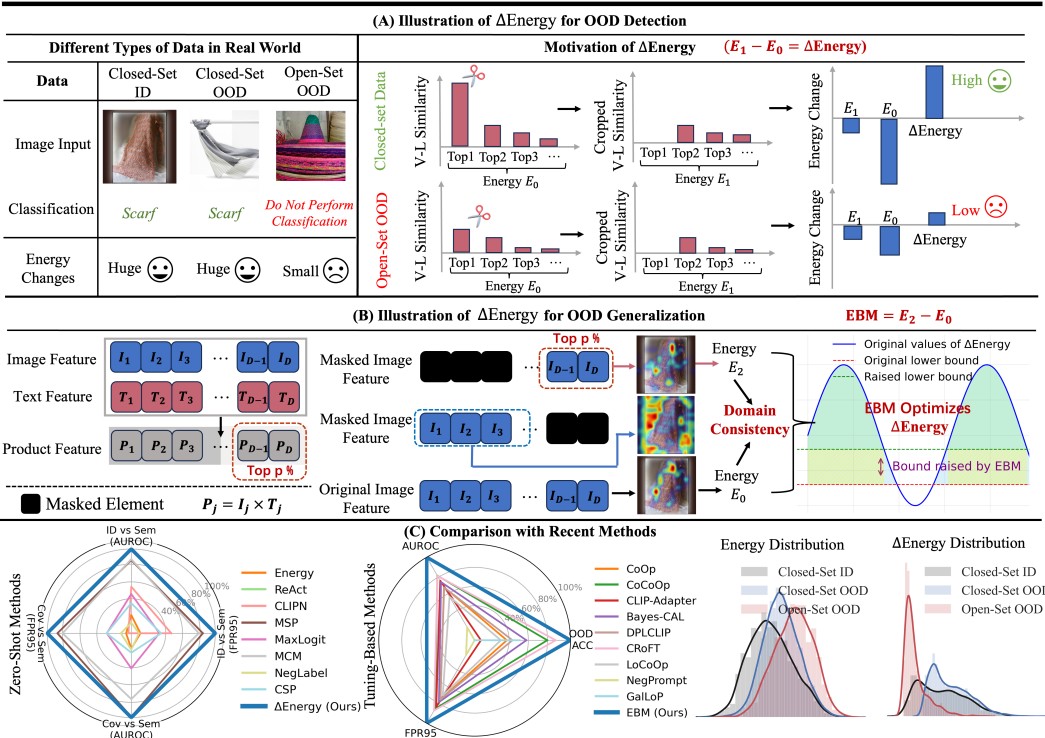

Figure 1: (A) Illustration of $\Delta$Energy for OOD detection. Significant differences in $\Delta$Energy are observed between closed-set data and open-set OOD data when the maximum cosine similarity is cropped to zero. (B) Illustration of the $\Delta$Energy for OOD generalization. We introduce the EBM method to achieve domain-consistent Hessians, which simultaneously triggers bound optimization for $\Delta$Energy. More details are in Section 3.2. (C) Comparison between our $\Delta$Energy and EBM with state-of-the-art methods. In the radar plots, all values are normalized to the range [0, 1]. It is observed that recent methods aimed at improving VLMs' OOD detection may not scale well to handling different types of distribution shifts in challenging ImageNet-1k OOD datasets.

classes (Wang et al., 2023b,a). Therefore, it is essential to *develop robust models that enhance VLMs' generalization ability to closed-set OOD data, while also effectively detecting open-set OOD classes during fine-tuning.*

However, most previous studies (Wortsman et al., 2022; Chen et al., 2024; Jiang et al., 2023; Goyal et al., 2023; Wang et al., 2023a; Ming et al., 2022a; Bai et al., 2024; Li et al., 2024a) have primarily focused on improving VLMs' robustness to training classes or developing OOD detection method for unseen classes independently. Consequently, existing approaches are often highly specialized for a single task and are not capable of simultaneously addressing both aspects. Recent works (Yang et al., 2023; Zhang et al., 2023a) have taken into account both shift types and introduced full-spectrum OOD (FS-OOD) detection, which considers both detecting semantic shifts and being tolerant to covariate shifts. While the FS-OOD benchmark evaluates OOD detection performance across various distribution types, it may not focus on improving VLM's classification accuracy on covariate-shifted data. Several studies (Lafon et al., 2024; Zhu et al., 2024, 2025a) also have attempted to tackle this issue using multiple diverse prompts or through energy optimization techniques. However, these approaches (Lafon et al., 2024; Zhu et al., 2024) often require significantly more computational resources to train additional local prompts or have been evaluated on a narrow set of post-hoc functions for OOD detection. Thus, when fine-tuning VLMs for downstream tasks, the challenge of improving the VLMs' generalization ability to closed-set OOD data while simultaneously detecting open-set OOD classes that were unseen during fine-tuning remains largely underexplored.

In this paper, we develop novel zero-shot and few-shot fine-tuning paradigms to go beyond the limitations of previous studies. We begin by proposing a new post-hoc OOD detection method, inspired by the following heuristic observation: Given the text prompts corresponding to ID data and input images, we compute the cosine similarities between the image features and text features.

As shown in Figure 1 (A), when we crop the maximum cosine similarity to a low value (such as by resetting to zero), the resulting change in energy score (Liu et al., 2020) is substantially different between closed-set data and open-set semantic-shifted classes.

> **Takeaways for $\Delta$Energy when aligning vision language modalities**
>
> When re-aligning vision-language modalities by setting the maximum cosine similarity to zero, we define the resulting change in energy score as $\Delta$Energy. As demonstrated in Theorem 3.2, the $\Delta$Energy for ID data is consistently larger than that for OOD data, indicating that it provides a discriminative and effective method for OOD detection. Meanwhile, compared to the MCM method and raw energy scores, $\Delta$Energy amplifies the difference between ID and OOD data—a property supported by both theoretical analysis and empirical evidence. Extensive experiments further demonstrate that our method outperforms state-of-the-art zero-shot OOD detection approaches on hard OOD detection benchmarks.

Building on this insight, we propose leveraging the energy change to distinguish closed-set classes from open-set OOD classes. We introduce a zero-shot OOD detection method, termed $\Delta$Energy, which quantifies the energy change resulting from modifying vision-language alignment (i.e., the cosine similarities). As demonstrated in Section 4, $\Delta$Energy significantly outperforms recent methods in detecting hard OOD classes, providing a more reliable approach for OOD detection.

Moreover, $\Delta$Energy can be further optimized to enhance OOD detection while simultaneously improving OOD generalization. This is achieved through $\Delta$**E**nergy-based **b**ound **m**aximization (termed EBM) during few-shot adaptation of VLMs. As depicted in Figure 1 (B), we modify the vision-language alignment by retaining the $p\%$ of the image feature elements (with $p$ as a hyperparameter) and masking the remaining elements. The resulting masked features are then used to compute a new energy change between the original and masked models, which we refer to as EBM. As demonstrated in Theorem 3.4, minimizing EBM is theoretically shown to maximize the lower bound of $\Delta$Energy. Moreover, the EBM method not only theoretically enhances the discrimination between closed-set known classes and open-set OOD classes based on the newly introduced OOD score (See Theorem 3.4), but also leads to stronger OOD generalization under covariate shifts (See Theorem 3.5). This allows us to fine-tune VLMs in a unified framework, enhancing both OOD generalization for closed-set OOD data and OOD detection for open-set OOD data.

## 2    Preliminary

In this section, we first provide the data setting in Notation 2.1 and formally define the target tasks. Based on the widely-used vision-language model CLIP (Radford et al., 2021), we then present the motivation for modifying the vision-language alignment through a masking operation.

**Notation 2.1.** Given the in-distribution (ID) samples from the downstream task, $\{\mathbf{x_i}, \mathbf{y_i}\}_{i=1}^{N}$, we define the classes of these samples as closed-set classes, while the other classes are considered as open-set OOD classes. The text prompts for the closed-set classes are defined as $\mathcal{T}_{\text{in}} = \{t_1, t_2, ..., t_K\}$, where $K$ represents the number of closed-set classes. Each text prompt $t_i$ can be formulated as "a photo of a {CLASS NAME}". Based on a pre-trained VLM, we can obtain the zero-shot image features and text features, denoted as $\{\mathbf{z_I}(\mathbf{x_i})\}_{i=1}^{N}$ and $\{\mathbf{z_T}(t_i)\}_{i=1}^{K}$, respectively. Both $\mathbf{z_I}(\mathbf{x_i})$ and $\mathbf{z_T}(t_i)$ are $D$-dimensional features.

**Task definition** Given a set of *closed-set ID* samples $\{\mathbf{x_i}, \mathbf{y_i}\}_{i=1}^{N}$, drawn from a source domain $\mathcal{S}$, the model is tasked with learning a robust predictor $f : \mathcal{X} \to \mathcal{Y}$, which maps inputs $\mathbf{x} \in \mathcal{X} = \mathbb{R}^{D_0}$ to outputs $\mathbf{y} \in \mathcal{Y} = \mathbb{R}^{K}$, where $D_0$ is the dimension of $\mathbf{x}$ and $K$ is the class number. Here, $N$ is the total number of the few-shot ID samples. To effectively address both closed-set OOD data (covariate shifts) and open-set OOD data (semantic shifts), we aim to enhance the robustness of predictor $f$ from two perspectives: 1) *OOD generalization*, which requires the model to generalize on closed-set classes from new domains $\mathcal{T}$ that exhibit covariate shifts; and 2) *OOD detection*, which enables the model to detect open-set OOD classes during test time.

**Effect of vision-language re-alignment through masking** We illustrate how vision-language re-alignment is achieved through a specific masking strategy during fine-tuning. Given an image input $\mathbf{x_i}$ and the text prompt $t$ that yields the maximum cosine similarity, we denote the corresponding

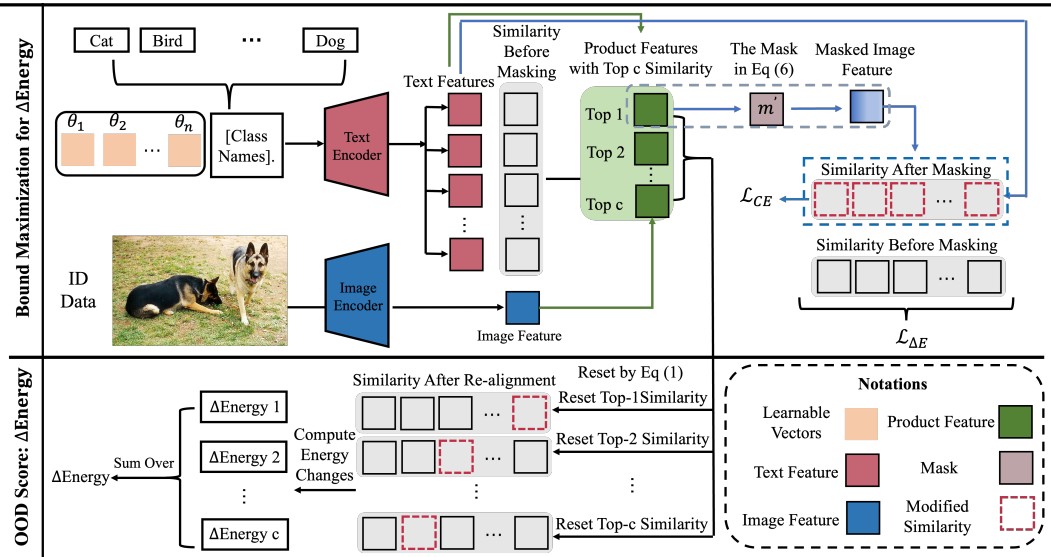

Figure 2: Overview of the proposed method. Based on the prompt-tuning approach, we freeze both the image encoder and the text encoder, making only the context vectors ($\theta = [\theta_1, \cdots, \theta_n]$) learnable under the proposed objective function, as shown in Equation 8. During fine-tuning, we apply a masking operation to each ID image feature based on the top-1 similarity, as defined in Equation 6. We then compute the resulting energy change after modifying the vision-language alignment via masking, which allows us to perform bound optimization on $\Delta$Energy. In the inference phase, following Equation 1, we reset the top-$c$ cosine similarities and then compute $\Delta$Energy for OOD detection. Simultaneously, we use the fine-tuned text feature and unmasked image feature for classification at test time. The complete algorithm can be seen in Appendix G.

zero-shot image feature and text feature as $\mathbf{z_I}(\mathbf{x_i})$ and $\mathbf{z_T}(t)$, respectively. We then compute their element-wise product, represented as $\mathbf{z_P}(\mathbf{x_i}) := \mathbf{z_I}(\mathbf{x_i}) \odot \mathbf{z_T}(t)$. Let $I_j$, $T_j$, and $P_j$ denote the $j$-th element of $\mathbf{z_I}(\mathbf{x_i})$, $\mathbf{z_T}(t)$, and $\mathbf{z_P}(\mathbf{x_i})$, respectively, such that $I_j \cdot T_j = P_j$. Based on the product vector $\mathbf{z_P}(\mathbf{x_i})$, we mask (zero-out) elements in $\mathbf{z_I}(\mathbf{x_i})$ where $P_j > 0$. *This masking operation thus effectively reduces the maximum cosine similarity to a low value, achieving modified vision-language alignment.* Moreover, from the attention visualization in Figure 1 (B), the pre-trained VLM initially focuses on the foreground object. However, after masking the elements of the image feature where $P_j > 0$, the model's attention shifts and becomes more reliant on background information. In contrast, masking the elements where $P_j < 0$ preserves the model's original attention, which motivates us to leverage this consistency between the original and masked domains to improve OOD generalization. Additional visualizations are provided in Figure 3 in Appendix G.

## 3 Methodology

Building upon the heuristic observations as shown in Figure 1, we propose a novel OOD score, named $\Delta$Energy, that measures the energy change when re-aligning vision-language modalities. We theoretically demonstrate that $\Delta$Energy outperforms the widely-used MCM (Ming et al., 2022a) (see Theorems 3.2-3.3). Moreover, we introduce a $\Delta$Energy-based bound maximization, which is proven to not only enhance OOD detection (see Theorem 3.4) but also lead to stronger OOD generalization (see Theorem 3.5). Before delving into the details, we provide the notations in this section as follows:

**Notation 3.1.** We define the cosine similarity [2] between the image feature $\mathbf{z_I}(\mathbf{x_i})$ and text feature $\mathbf{z_T}(t_j)$ as $s_j(\mathbf{x_i}) = \mathbf{z_I}(\mathbf{x_i}) \cdot \mathbf{z_T}(t_j)$. Let $\hat{y}_1 := \arg\max_{i \in [K]} s_i(\mathbf{x_i})$ denote the index of the maximum cosine similarity and $\hat{y}_j := \arg\max_{i \in [K] \setminus \{\hat{y}_1, \cdots, \hat{y}_{j-1}\}} s_i(\mathbf{x_i})$ denote the index of the $j$-th largest cosine similarity. And we denote the corresponding text features that have the $j$-th largest similarity with $\mathbf{z_I}(\mathbf{x_i})$, as $\mathbf{h_j}(\mathbf{x_i}) := \mathbf{z_T}(\hat{t}_j(\mathbf{x_i}))$. Here, $\hat{t}_j(\mathbf{x_i})$ refers to the text prompt corresponding to the $j$-th largest cosine similarity.

---

[2]In this paper, $\mathbf{z_I}(\mathbf{x_i})$ and $\mathbf{z_T}(t_j)$ are normalized features

## 3.1 ΔEnergy for OOD detection

The proposed $\Delta$Energy, which measures the energy change after modifying the top-$c$ maximum cosine similarities [3], unfolds as follows:

- Based on a pre-trained VLM, for each image feature $\mathbf{z_I}(\mathbf{x_i})$, we first select the text feature sets $\{\mathbf{h_j}(\mathbf{x_i})\}_{j=1}^c$ that have the top $c$ similarity with $\mathbf{z_I}(\mathbf{x_i})$.

- We then compute the product between each image feature $\mathbf{z_I}(\mathbf{x_i})$ and the selected text feature $\mathbf{h_j}(\mathbf{x_i})$. The product feature is represented as $\mathbf{z_P}(\mathbf{x_i}, \hat{t}_j) = \mathbf{z_I}(\mathbf{x_i}) \odot \mathbf{h_j}(\mathbf{x_i})$.

- For each text feature $\mathbf{h_j}(\mathbf{x_i})$ ($j \in [1, \cdots, c]$), we denote the $j$-th largest cosine similarity between the image feature and the text feature as $s_{\hat{y}_j}(\mathbf{x_i}) = \mathbf{z_I}(\mathbf{x_i}) \cdot \mathbf{h_j}(\mathbf{x_i})$. Let $\tilde{s}_{\hat{y}_j}(\mathbf{x_i})$ represents the new cosine similarity after re-alignment, which is achieved by:

$$\tilde{s}_{\hat{y}_j}(\mathbf{x_i}) = 0 \tag{1}$$

- Finally, we can compute the new OOD score as: $\Delta\text{Energy}(\mathbf{x_i}) = E_1(\mathbf{x_i}) - E_0(\mathbf{x_i})$. Based on the scaling temperature $\tau$, $E_0(\mathbf{x_i})$ is the energy score before the re-alignment:

$$E_0(\mathbf{x_i}) = -\log \sum_{j=1}^K e^{s_j(\mathbf{x_i})/\tau} \tag{2}$$

$E_1(\mathbf{x_i})$ is the energy score after the re-alignment:

$$E_1(\mathbf{x_i}) = -\frac{1}{c} \sum_{j=1}^c \log \left[ e^{\tilde{s}_{\hat{y}_j}(\mathbf{x_i})/\tau} + \sum_{p \neq \hat{y}_j} e^{s_p(\mathbf{x_i})/\tau} \right] \tag{3}$$

We provide formal guarantees that the proposed $\Delta$Energy can provably surpass the widely-used VLM-based OOD detection method MCM (Ming et al., 2022a).

**Theorem 3.2.** *[OOD Detection Ability of* $\Delta$**Energy***] Suppose that the maximum cosine similarity for an ID sample* $\mathbf{x}_{ID}$ *is greater than that of an open-set OOD sample* $\mathbf{x}_{OOD}$*, i.e.,* $s_{\hat{y}_1}(\mathbf{x}_{ID}) > s_{\hat{y}_1}(\mathbf{x}_{OOD})$*. Let* $S_{\text{Method}}(\mathbf{x})$ *denote the score assigned to sample* $\mathbf{x}$ *under a given method. We have the following properties: 1)* $S_{\Delta\text{Energy}}(\mathbf{x}_{ID}) > S_{\Delta\text{Energy}}(\mathbf{x}_{OOD})$ *for ID (*$\mathbf{x}_{ID}$*) and open-set OOD (*$\mathbf{x}_{OOD}$*) samples. 2) Compared to the MCM method,* $\Delta$Energy *amplifies the difference between ID and OOD data, i.e.,* $d_{\Delta\text{Energy}} > d_{MCM}$*, where* $d_{\text{Method}} = S_{\text{Method}}(\mathbf{x}_{ID}) - S_{\text{Method}}(\mathbf{x}_{OOD})$*.*

**Theorem 3.3.** *[The proposed OOD Score* $\Delta$**Energy** *gets lower FPR than MCM] Given a task with closed-set ID label set* $\mathcal{Y}_{\text{in}} = \{y_1, y_2, ..., y_K\}$ *and a pre-trained VLM, for any test input* $\mathbf{x}'$*, based on the scaling temperature* $\tau$*, the maximum concept matching (MCM) score is computed as follows:*

$$S_{\mathbf{MCM}}(\mathbf{x}'; \mathcal{Y}_{\text{in}}) = \max_i \frac{e^{s_i(\mathbf{x}')/\tau}}{\sum_{j=1}^K e^{s_j(\mathbf{x}')/\tau}}.$$

*For any* $c \in \{1, 2, \cdots, K\}$*, if* $s_{\hat{y}_1}(\mathbf{x}') \leq \tau \ln 2$*, we have*

$$\text{FPR}^{\Delta\text{Energy}}(\tau, \lambda) \leq \text{FPR}^{\text{MCM}}(\tau, \lambda),$$

*where* $\text{FPR}^{\Delta\text{Energy}}(\tau, \lambda)$ *and* $\text{FPR}^{\text{MCM}}(\tau, \lambda)$ *is the false positive rate of* $\Delta$Energy *and* MCM*, respectively, based on the temperature* $\tau$ *and detection threshold* $\lambda$*.*

## 3.2 The ΔEnergy-based bound maximization enhances OOD detection and generalization

Furthermore, as illustrated in Equation 6, we introduce a $\Delta$Energy-based bound maximization function (EBM) during the fine-tuning process, which aims at increasing the lower bound of $\Delta$Energy score for closed-set classes as demonstrated in Theorem 3.4. As illustrated in Theorem 3.5, the proposed objective function is theoretically proven to not only improve OOD detection but also lead to a domain-consistent Hessian, which serves as a strong indicator of OOD generalization.

---

[3]To enlarge the energy change for closed-set data, we perform re-alignment based on the top-$c$ similarities.

Specifically, motivated by further enlarging $\Delta$Energy, we propose to minimize the following term:

$$\mathcal{L}_{\Delta E} = \frac{1}{N} \sum_{i=1}^{N} [E_2(\mathbf{x_i}) - E_0(\mathbf{x_i})] \tag{4}$$

where $N$ is the number of few-shot ID samples during fine-tuning and $E_2(\mathbf{x_i})$ is the energy score for $\mathbf{x_i}$ after masking on the image feature, which is formally calculated as:

$$E_2(\mathbf{x_i}) = -\log \sum_{j=1}^{K} e^{s'_j(\mathbf{x_i})/\tau} \tag{5}$$

$$s'_j(\mathbf{x_i}) = (\mathbf{z_I}(\mathbf{x_i}) \odot \mathbf{m}'(\mathbf{x_i})) \cdot \mathbf{z_T}(t_j) \tag{6}$$

Here, $\mathbf{m}'(\mathbf{x_i})$ is the mask that retains the top $p$-proportion elements in $\mathbf{z_I}(\mathbf{x_i}) \odot \mathbf{h_1}(\mathbf{x_i})$ and $\mathbf{h_1}(\mathbf{x_i})$ is the text feature corresponding to the top-1 cosine similarity.

**Theorem 3.4.** *[EBM increase the lower bound of $\Delta$Energy] Let $\mathbf{h_1}(\mathbf{x_i})$ denote the the text feature that have the top-1 similarity with the image feature $\mathbf{z_I}(\mathbf{x_i})$. The corresponding similarity is computed as $s_{\hat{y}_1}(\mathbf{x_i}) = \mathbf{z_I}(\mathbf{x_i}) \cdot \mathbf{h_1}(\mathbf{x_i})$. Suppose that $\mathcal{L}_{\Delta E} \leq \varepsilon_E$, with $c = 1$, under the condition that:*

$$\sum_{i=1}^{N} e^{s_{\hat{y}_1}(\mathbf{x_i})/\tau} - e^{\tilde{s}_{\hat{y}_1}(\mathbf{x_i})/\tau} \geq \sum_{i=1}^{N} (e^{\varepsilon_E} - 1) e^{-E_2(\mathbf{x_i})} \tag{7}$$

*we have $\frac{1}{N} \sum_{i=1}^{N} \Delta\text{Energy}(\mathbf{x_i}) \geq -\mathcal{L}_{\Delta E}$.*

Theorem 3.4 implies that if the change in the VLM's predictions after re-alignment is not too small and satisfies the condition as shown in Equation 7, minimizing $\mathcal{L}_{\Delta E}$ can increase the lower bound of $\Delta$Energy for closed-set classes. Moreover, we theoretically demonstrated that the proposed EBM loss in Equation 6 can also lead to domain-consistent Hessians of classification loss, which serves as a strong indicator for OOD generalization (Rame et al., 2022; Hemati et al., 2023).

**Theorem 3.5.** *[EBM leads to domain-consistent Hessians] Given the ID training data sampled from domain $\mathcal{S}$ and the learnable parameter $\theta$ in VLM, we denote the masked domain as $\mathcal{S}'$. We represent the empirical classification loss on the domain $\mathcal{D}$ as $\widehat{\mathcal{E}}_{\mathcal{D}}(\theta)$. Let $\widehat{\mathbf{G}}_{\mathcal{D}}(\theta)$ and $\widehat{\mathbf{H}}_{\mathcal{D}}(\theta)$ be the gradient vector and Hessian matrix of empirical risk $\widehat{\mathcal{E}}_{\mathcal{D}}(\theta)$ over parameter $\theta$, respectively. In this paper, we propose to minimize $\mathcal{L}_{\Delta E}$. The distance between the unmasked and masked image feature is assumed to satisfy: $\|\mathbf{z_I}(\mathbf{x_i}) - (\mathbf{z_I}(\mathbf{x_i}) \odot \mathbf{m}'(\mathbf{x_i}))\|_2 \leq \varepsilon$. Then the local optimum $\theta$ of $\min \mathcal{L}_{\Delta E}$ satisfies:*

$$|\theta^{\top}(\widehat{\mathbf{H}}_{\mathcal{S}}(\theta) - \widehat{\mathbf{H}}_{\mathcal{S}'}(\theta))\theta| \leq \frac{\varepsilon}{N} \sum_{i=1}^{N} |\theta^{\top} \nabla_{\theta}^2 \mathbf{z_T}(\mathbf{x_i})\theta|$$

**Proposition 3.6.** *[EBM bound OOD generalization] Let $\mathbf{z_I}(\mathbf{x_i})$ and $\tilde{\mathbf{z}}_{\mathbf{I}}(\mathbf{x_i})$ denote the image feature from source domain ($\mathcal{S}$) and target domain ($\mathcal{T}$), respectively. We assume that $\|\mathbf{z_I}(\mathbf{x_i}) - \tilde{\mathbf{z}}_{\mathbf{I}}(\mathbf{x_i})\|_2 \leq \varepsilon_1$. By applying the second-order Taylor expansion and utilizing the domain-consistent Hessians as outlined in Theorem 3.5, the OOD generalization gap between source domain ($\mathcal{S}$) and target domain ($\mathcal{T}$) is upper bounded by the following inequality:*

$$\max_{\{\theta : |\widehat{\mathcal{E}}_{\mathcal{S}}(\theta) - \widehat{\mathcal{E}}_{\mathcal{S}}(\theta^*)| \leq \epsilon\}} |\widehat{\mathcal{E}}_{\mathcal{T}}(\theta) - \widehat{\mathcal{E}}_{\mathcal{S}}(\theta^*)| \lesssim |\widehat{\mathcal{E}}_{\mathcal{T}}(\theta^*) - \widehat{\mathcal{E}}_{\mathcal{S}}(\theta^*)| + \max \frac{1}{2} |\theta^{\top} \widehat{\mathbf{H}}_{\mathcal{S}}(\theta^*)\theta| + O(\varepsilon_1)$$

*where $\theta^*$ is a local minimum across all domains, i.e., $\nabla_{\theta} \widehat{\mathcal{E}}_{\mathcal{D}}(\theta^*) = \mathbf{0}$.*

Therefore, by connecting the EBM loss with the Hessians of empirical classification loss, we theoretically discover that the EBM loss can lead to a bound of the performance gap between closed-set ID data and closed-set OOD data. This implies that optimizing for $\Delta$Energy with EBM loss also involves optimizing for OOD generalization.

Table 1: **OOD detection between closed-set data and open-set OOD data based on ImageNet-1k:** OOD detection measured by AUROC and FPR95 over the mixture of closed-set test sets and open-set OOD test sets.

| DATA | Method | Energy | ODIN | ReAct | CLIPN | MSP | MaxLogit | MCM | NegLabel | CSP | ΔEnergy (Ours) |
|---|---|---|---|---|---|---|---|---|---|---|---|
| ID vs. | FPR95↓ | 76.72 | 51.71 | 80.38 | 64.37 | 51.72 | 69.12 | 53.34 | 77.31 | 68.49 | **46.40** (2.58) |
| Semantic-shifted OOD | AUROC↑ | 76.94 | 85.61 | 74.06 | 81.44 | 85.64 | 80.28 | 85.81 | 75.73 | 78.84 | **87.10** (0.75) |
| Covariate-shifted OOD vs. | FPR95↓ | 83.94 | **61.25** | 85.05 | 84.44 | 69.20 | 79.80 | 70.30 | 82.81 | 79.76 | 67.16 (0.38) |
| Semantic-shifted OOD | AUROC↑ | 67.21 | 77.10 | 64.66 | 64.64 | 78.64 | 70.52 | 75.66 | 67.38 | 67.89 | **78.68** (0.57) |

Table 2: **OOD detection between closed-set data and open-set OOD data based on PACS and VLCS:** OOD detection measured by AUROC and FPR95 over the mixture of closed-set OOD and open-set OOD test sets.

| DATA | PACS vs. Open-Set (AUC↑ / FPR95↓) | | | VLCS vs. Open-Set (AUC↑ / FPR95↓) | | | AVG |
|---|---|---|---|---|---|---|---|
| Method | DTD | Food101 | Caltech101 | DTD | Food101 | Caltech101 | AUC↑ / FPR95↓ |
| Energy | 82.4 / 67.6 | 95.9 / 26.0 | 86.7 / 52.2 | 55.3 / 88.8 | 85.8 / 48.3 | 53.3 / 86.3 | 76.6 / 61.5 |
| ReAct | 89.5 / 44.9 | 98.1 / 9.9 | 89.8 / 43.2 | 52.8 / 89.7 | 86.7 / 47.8 | 61.4 / 83.2 | 79.7 / 53.1 |
| CLIPN | 93.6 / 40.3 | 96.2 / 25.6 | 88.1 / 56.1 | 80.4 / 62.7 | 88.9 / 46.6 | 72.5 / 74.1 | 86.6 / 50.9 |
| MaxLogit | 89.5 / 45.1 | 97.9 / 13.1 | 88.9 / 46.8 | 60.8 / 87.6 | 88.9 / 45.4 | 69.4 / 81.6 | 82.6 / 53.3 |
| MSP | 97.9 / 9.8 | 98.9 / 4.6 | 95.8 / 20.9 | 84.4 / 55.3 | 93.7 / 35.1 | 88.9 / 48.8 | 93.3 / 29.1 |
| ODIN | 99.1 / 1.8 | 99.3 / 1.0 | 97.4 / 6.7 | 83.1 / 46.5 | 91.3 / 28.2 | 86.8 / 40.9 | 92.8 / **17.9** |
| MCM | 98.9 / 4.3 | 99.2 / 3.4 | 97.0 / 13.7 | 84.2 / 55.1 | 93.3 / 36.4 | 88.5 / 50.4 | 93.5 / 27.2 |
| NegLabel | 99.3 / 4.2 | 97.7 / 15.8 | 95.7 / 31.6 | 84.8 / 54.3 | 79.4 / 74.6 | 62.7 / 84.3 | 86.6 / 49.1 |
| CSP | 99.6 / 1.8 | 99.2 / 2.6 | 97.8 / 12.4 | 88.9 / 38.7 | 78.3 / 67.4 | 67.3 / 73.0 | 88.5 / 32.7 |
| ΔEnergy (Ours) | 98.1 / 6.5 | 99.2 / 2.4 | 96.1 / 14.3 | 85.3 / 53.2 | 94.1 / 31.9 | 89.5 / 47.3 | **93.7** / 25.9 |

## 3.3 Overview of the proposed method

Our theoretical analysis thus leads to the design of a new fine-tuning framework with concurrent optimization for both tasks. As illustrated in Figure 2, we prioritize computational efficiency by adopting prompt-tuning techniques. In the fine-tuning process, both the image encoder and text encoder are frozen and only the context vectors $\theta$ are learnable. Let $\mathcal{L}_{CE}$ denote the Cross-Entropy loss and $\lambda_0$ denote the hyperparameter that can be chosen based on the validation procedure. Then the final optimization objective of the EBM method is expressed as:

$$\mathcal{L}_{EBM} = \mathcal{L}_{CE} + \lambda_0 e^{\mathcal{L}_{\Delta E}} \tag{8}$$

## 4 Experiments

In this section, motivated by the remarkable success of the vision-language model CLIP (Radford et al., 2021) in learning general visual knowledge, we conduct experiments based on CLIP. First, we conduct extensive experiments to validate the effectiveness of the proposed $\Delta\mathrm{Energy}$ in zero-shot OOD detection across various datasets. Furthermore, we evaluate the effect of the proposed EBM method in enhancing both OOD generalization and OOD detection. Due to space limitations, we provide ablation studies in Appendix G to validate our theoretical findings.

### 4.1 Effectiveness of $\Delta\mathrm{Energy}$ for OOD detection

**Dataset** We evaluate OOD detection performance over 4 different benchmarks, including 1) the discrimination between closed-set OOD data and open-set OOD data based on ImageNet-1k, 2) the discrimination between closed-set OOD data and open-set OOD data based on cross-dataset images, 3) hard OOD detection on different splits of ImageNet-1k, 4) the conventional OOD detection benchmark. For the first two benchmark datasets, we evaluate the models' OOD detection capabilities under more challenging scenarios, where the datasets exhibit both covariate and semantic shifts. Models are required to distinguish between various types of closed-set OOD data (covariate shifts) and open-set OOD data (semantic shifts). Details of the two data settings are illustrated as follows:

1) *Setup-I: open-set discrimination on the large-scale ImageNet-1k dataset.* Following the prior work (Zhu et al., 2024), we split ImageNet-1k (Krizhevsky et al., 2017) into open and closed sets w.r.t class labels. We randomly define 40% classes of ImageNet as the closed-set, and the remaining 60% as the open-set. The samples from ImageNet-A (Hendrycks et al., 2021b), ImageNet-R (Hendrycks et al., 2021a), ImageNet-Sketch (Wang et al., 2019), and ImageNet-V2 (Recht et al., 2019), which share the same class labels as the closed-set ID data, are considered as closed-set OOD data.

2) *Setup-II: open-set discrimination on cross-dataset images.* Using cross-dataset examples as the open-set is another established protocol (Shafaei et al., 2018; Kong and Ramanan, 2021). Following

Table 3: **Tuning-based results on Setup-I:** Comparison with competitive fine-tuning methods based on CLIP ViT-B/16 using 16 samples per class. In the testing phase of LoCoOp, NegPrompt and GalLoP, we use the GL-MCM score (Miyai et al., 2023) to compute OOD detection results.

| Algorithm OOD Score | CoOp MCM | CoCoOp MCM | CLIP-Adapter MCM | Bayes-CAL MCM | DPLCLIP MCM | CRoFT MCM | LoCoOp GL | NegPrompt GL | GalLoP GL | **EBM (Ours)** $\Delta$Energy |
|---|---|---|---|---|---|---|---|---|---|---|
| ID ACC ↑ | 82.11 | 81.59 | 79.91 | 82.31 | 82.46 | 82.03 | 82.14 | 81.46 | **84.51** | 81.52 (0.4) |
| OOD ACC ↑ | 61.36 | 62.58 | 60.58 | 61.95 | 61.53 | 62.83 | 61.18 | 60.39 | 61.75 | **63.28 (0.2)** |
| AUROC ↑ | 72.94 | 76.38 | 74.86 | 74.44 | 72.81 | 76.30 | 70.03 | 60.86 | 56.97 | **81.90 (1.9)** |
| FPR95 ↓ | 73.15 | 70.30 | 70.92 | 72.34 | 73.07 | 69.78 | 74.33 | 86.66 | 91.17 | **65.90 (1.7)** |

the prior work (Gulrajani and Lopez-Paz, 2021; Cha et al., 2022; Ye et al., 2021), we leverage popular datasets like PACS (Li et al., 2017) or VLCS (Li et al., 2017) for domain generalization studies as the closed-set data. We evaluate the models' ability to distinguish between closed-set OOD and cross-dataset images by utilizing different styles of datasets like Caltech101 (Bansal et al., 2021), DTD (Sharan et al., 2014), and Food101 (Bossard et al., 2014) as open-set OOD examples. All overlapping classes are removed from the three open-set OOD datasets.

In addressing the hard OOD detection scenarios, we follow the prior works (Ming et al., 2022a; Li et al., 2024a; Chen et al., 2024) and partition the ImageNet1k dataset into two parts: one part of the data serves as the ID, while the other serves as OOD. For conventional OOD detection, we use a popular benchmark in which ImageNet-1k (Krizhevsky et al., 2017) with 1,000 classes is used as the ID dataset, and the OOD datasets including subsets of Texture (Cimpoi et al., 2014), iNaturalist (Van Horn et al., 2018), Places (Zhou et al., 2017) and SUN (Xiao et al., 2010).

**Comparison methods** To substantiate the effectiveness of the proposed OOD score, we conduct an empirical analysis of distinct categories of methodologies for OOD detection utilizing VLMs. These categories encompass zero-shot approaches and the methods that combine the CLIP image encoder with classical approaches. For zero-shot methods, we opted for 4 recent methods, MCM (Ming et al., 2022a), CLIPN (Wang et al., 2023a), NegLabel (Jiang et al., 2024) and CSP (Chen et al., 2024). MCM employs the original CLIP, utilizing the maximum softmax probability operation on the similarities for detection, and CLIPN involves an additional training phase during pre-training, specifically training a negative text encoder using large external data. Both the NegLabel and CSP methods introduce additional negative labels, enabling more accurate OOD detection. For the second group of methods, we adapt previous logits-based methodologies to the use of the CLIP image encoder, including MSP (Hendrycks and Gimpel, 2016), Energy (Liu et al., 2020), MaxLogit (Hendrycks et al., 2019), ReAct (Sun et al., 2021) and ODIN (Liang et al., 2017a). Following the previous studies (Wang et al., 2023a), we use CLIP based on ViT-B/16, which is pre-trained from OpenCLIP.

**Metrics** Two OOD detection metrics are used. The first is the False Positive Rate at a 95% True Negative Rate (FPR95), which denotes the rate of falsely identified OOD instances when the true negative rate is maintained at 95%. The second is the Area Under the Receiver Operating Characteristic curve (AUROC), representing the measure of OOD ranking across various classification thresholds.

**Experiments results** For zero-shot OOD detection, we set $c = 2$ and $\tau = 0.01$ in our $\Delta$Energy. We present the results of $\Delta$Energy and competitors in discriminating between closed-set data and open-set OOD data in Table 1-2. Due to space limitations, we provide the results on the traditional OOD detection and hard OOD detection datasets in Table 5-7 in Appendix G. It is observed that the proposed $\Delta$Energy obtains the top-1 AUROC performance on all benchmarks. Compared with the vanilla energy-based OOD detection method (Liu et al., 2020), our $\Delta$Energy method consistently achieves better OOD detection performance across all 4 benchmarks. Notably, as shown in Table 1–2, our approach surpasses the competitive NegLabel and CSP methods by a large margin, demonstrating the superiority of the proposed method in distinguishing different semantics in open-world scenarios. The inferior performances of NegLabel and CSP may stem from the underlying assumption of these methods—that OOD samples possess a variety of distinct visual properties—which may not hold in hard OOD detection scenarios where OOD samples are distributed closely with closed-set data. Furthermore, the presence of covariate shifts reduces the similarity between closed-set OOD and ID data, thereby making it more difficult to distinguish closed-set OOD data from open-set OOD data.

### 4.2 Effectiveness of the EBM loss for both OOD generalization and OOD detection

**Datasets** To substantiate the effectiveness of the proposed EBM method for both tasks, we evaluate our method under two data settings (Setup-I and Setup-II), each incorporating both covariate shifts and semantic shifts, as introduced in Section 4.1. For evaluating OOD generalization performance on

Table 4: **Tuning-based results on Setup-II:** Comparison with competitive fine-tuning methods based on CLIP ViT-B/16 using 16 samples per class.

| DATA | PACS | PACS vs. Open-Set (AUROC↑ / FPR95↓) | | | VLCS | VLCS vs. Open-Set (AUROC↑ / FPR95↓) | | | AVG |
|---|---|---|---|---|---|---|---|---|---|
| Algorithm | OOD ACC↑ | DTD | Food101 | Caltech101 | OOD ACC↑ | DTD | Food101 | Caltech101 | FPR↓ |
| ZS CLIP+ MCM | 96.1 (0.0) | 98.9 / 4.3 | 99.2 / 3.4 | 97.0 / 13.7 | 75.1 (0.0) | 84.2 / 55.1 | 93.3 / 36.4 | 88.5 / 50.4 | 27.2 |
| ZS CLIP + $\Delta$Energy | 96.1 (0.0) | 98.1 / 6.5 | 99.2 / 2.4 | 96.1 / 14.3 | 75.1 (0.0) | 85.3 / 53.2 | 94.1 / 29.7 | 89.5 / 47.3 | 25.5 |
| CoOp | 96.3 (0.7) | 98.9 / 4.6 | 99.2 / 3.1 | 97.4 / 11.2 | 78.3 (1.7) | 89.3 / 37.6 | 91.4 / 40.1 | 85.9 / 47.4 | 24.0 |
| CoCoOp | 96.8 (0.5) | 98.8 / 4.0 | 98.7 / 5.8 | 97.5 / 10.9 | 78.9 (0.8) | 88.3 / 44.9 | 89.2 / 44.6 | 86.8 / 48.6 | 26.5 |
| CLIP-Adapter | 96.1 (0.0) | 99.0 / 4.1 | 99.2 / 3.6 | 97.4 / 12.1 | 77.3 (0.7) | 85.9 / 52.4 | 93.8 / 35.5 | 89.3 / 48.5 | 26.0 |
| Bayes-CAL | 96.6 (0.5) | 98.5 / 7.2 | 98.3 / 8.8 | 95.9 / 16.6 | 79.6 (0.9) | 88.0 / 47.7 | 84.9 / 61.1 | 84.8 / 56.7 | 33.0 |
| DPLCLIP | 95.6 (0.2) | 96.6 / 21.6 | 97.1 / 18.1 | 92.2 / 35.1 | 76.5 (1.3) | 88.9 / 37.1 | 86.8 / 43.2 | 84.2 / 50.2 | 34.2 |
| LoCoOp | 96.5 (0.3) | 98.1 / 9.7 | 98.4 / 8.6 | 95.9 / 19.4 | 76.3 (0.7) | 86.1 / 50.9 | 84.3 / 62.8 | 83.3 / 59.7 | 35.2 |
| GalLop | 96.9 (0.2) | 98.6 / 5.6 | 98.3 / 8.9 | 95.4 / 18.2 | 81.3 (0.9) | 87.4 / 39.0 | 91.0 / 43.4 | 80.7 / 59.1 | 29.0 |
| CRoFT | **97.3 (0.1)** | 94.0 / 33.0 | 89.9 / 57.7 | 84.0 / 71.2 | 80.2 (1.0) | 90.2 / 40.6 | 80.1 / 70.3 | 79.0 / 66.4 | 56.5 |
| NegPrompt | 97.1(0.4) | 88.9 / 42.2 | 97.5 / 15.2 | 94.4 / 22.6 | 77.8 (0.6) | 93.4 / 29.6 | 89.8 / 39.6 | 75.2 / 72.4 | 36.9 |
| EBM + MCM | 97.2 (0.1) | 98.6 / 5.6 | 99.0 / 4.3 | 96.8 / 13.7 | **81.7 (0.6)** | 91.6 / 32.6 | 93.3 / 32.7 | 86.5 / 44.1 | 22.2 |
| EBM + $\Delta$Energy | 97.2 (0.1) | 98.2 / 6.3 | 98.7 / 3.8 | 96.6 / 14.8 | **81.7 (0.6)** | 91.7 / 30.2 | 93.6 / 28.6 | 86.4 / 42.5 | **21.0** |

Setup-II, we utilize the leave-one-domain-out validation protocol (Gulrajani and Lopez-Paz, 2020) that uses three domains as closed-set ID data and the remaining one as closed-set OOD data.

**Comparison methods** We conduct an empirical analysis of distinct categories of CLIP-based lightweight fine-tuning methods. In addition to comparing our approach with popular fine-tuning techniques, such as the widely-used CoOp (Zhou et al., 2021), CoCoOp (Zhou et al., 2022) and CLIP-Adapter (Gao et al., 2023), we also evaluate it against CLIP-based methods designed specifically for OOD generalization or OOD detection. For OOD generalization, we compare our EBM with methods like DPLCLIP (Zhang et al., 2021b) and Bayes-CAL (Zhu et al., 2023b, 2025b). For OOD detection, we consider approaches such as LoCoOp (Miyai et al., 2024b), NegPrompt (Li et al., 2024a), CRoFT (Zhu et al., 2024), and GalLoP (Lafon et al., 2024), with CRoFT and GalLoP explicitly designed to optimize both OOD generalization and detection.

**Experiment details** We conduct experiments based on the CLIP ViT-B/16 model using 16 samples per ID classes. For the prompt learning methods, we use random initialization for context vectors and set the number of context tokens to 16. Without otherwise specified, methods are trained using the SGD optimizer with a learning rate of 0.002 and batch size of 32 for fair comparisons. We set the maximum training epoch to 30 for all models. For all baseline methods, we follow the hyperparameter searching protocols recommended in their original papers. For our EBM method, we search for $\lambda_0$ in $[0.1, 0.5, 1.0]$ for Setup-I and we set $\lambda_0 = 2$ for Setup-II. We set $\tau = 0.01$ and $c = 2$ in $\Delta$Energy, and we vary the masking proportion $p\%$ within the range of $[0.4, 0.5, 0.6]$ in $\mathcal{L}_{\Delta E}$. For experiments on each method, we repeat 3 times with different random splits to eliminate the effects of randomness. Finally, we report the average classification accuracy on closed-set test sets, as well as the average FPR95 and AUROC results for distinguishing between open-set OOD data and closed-set OOD data based on MCM (GL-MCM) (Ming et al., 2022a; Miyai et al., 2023) or our $\Delta$Energy.

**Experiment results** We present the results of setup-I in Table 3, where the proposed EBM method establishes the overall best performance in both OOD generalization and OOD detection. Notably, our method outperforms the competitive CRoFT (Zhu et al., 2024) method, which is introduced to achieve concurrent optimization on both tasks. Our method obtains a 0.45% improvement on OOD accuracy in closed-set OOD data and more than 5% improvements on AUROC when discriminating closed-set OOD datasets and open-set OOD classes. In contrast, although the competitors can achieve higher ID test accuracy, they often struggle with the OOD generalization or OOD detection task, even resulting in worse performance when compared to the zero-shot CLIP model. Moreover, it is observed that recent methods, such as GalLoP (Lafon et al., 2024) and NegPrompt (Li et al., 2024a), aimed at improving VLMs' OOD detection may not scale well to handling different types of distribution shifts in large-scale ImageNet-1k datasets. The results of setup-II are shown in Table 4, where the EBM method also demonstrates the best overall performance on both tasks. Consistent with the results of Setup-I, its competitors, especially those (Zhu et al., 2023b; Zhang et al., 2021b; Zhu et al., 2024) designed for OOD generalization, achieve even higher FPR95 scores compared to the CLIP model. Notably, for the more challenging VLCS dataset, our EBM method shows significant performance improvements over CRoFT and GaLLoP on both tasks. This is likely due to the fact that CRoFT performs adapter-tuning on both the image and text inputs, which can lead to forgetting the general knowledge encoded in the pre-trained CLIP model. In contrast, based on fine-tuning language model, our method optimizes the energy change when re-aligning vision-language modalities, resulting in more robust performance across various datasets. These results highlight the EBM method's ability to more effectively enhance both OOD generalization and OOD detection.

# 5 Conclusions

Different from the vanilla energy-based OOD score, we propose a novel zero-shot OOD detection method, $\Delta\text{Energy}$, which measures the energy change when re-aligning vision-language modalities. Both theoretical and experimental results demonstrate that $\Delta\text{Energy}$ provides more reliable OOD detection than previous methods. Furthermore, we introduce a $\Delta\text{Energy}$-based bound maximization during fine-tuning VLMs. The proposed bound maximization is theoretically proven to not only improve OOD detection but also lead to optimization for OOD generalization. Building on this insight, we have developed a unified fine-tuning framework that enables the concurrent optimization of both tasks. Extensive experiments on challenging OOD detection and generalization benchmarks demonstrate the superiority of our method.

## Acknowledgements

This work is supported by National Natural Science Foundation of China (No.62572313, No.62106139).

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

# A    Related Works

**Robust fine-tuning methods for VLM** For training efficiency, there have been many lightweight CLIP-based fine-tuning methods to enhance generalization performance via prompt tuning (Singha et al., 2023; Huang et al., 2022; Khattak et al., 2023a; Wang et al., 2023c; Wasim et al., 2023; Goswami et al., 2024; Huang et al., 2024) or adapter tuning (Gondal et al., 2024; Zhang et al., 2023b; Song et al., 2023). Prompt tuning methods aim to get better vision-language alignment via only fine-tuning the input prompts. For example, with only few-shot samples for learning, CoOp (Zhou et al., 2021) improved significantly in generalization ability over intensively-tuned manual prompts via prompt learning. Motivated by learning generalization prompts, CoCoOp (Zhou et al., 2022) is proposed to achieve generalization on unseen classes via conditional prompt learning. Adapter-tuning is another popular lightweight fine-tuning method, like CLIP-Adapter (Gao et al., 2023), Tip-Adapter (Zhang et al., 2021a). Both of them inject a lightweight bottleneck architecture after the image encoder or text encoder and perform residual-style feature blending with the original pre-trained embeddings. However, most previous studies have focused on improving models' robustness to covariate shifts, without being able to effectively address OOD detection.

Note that several studies, such as CoCoOp (Zhou et al., 2022), PromptSRC (Khattak et al., 2023b), SHIP (Wang et al., 2023c), and Promptkd (Li et al., 2024b), have explored the base-to-new ability of VLMs, aiming to improve VLMs' classification performance to unseen classes. However, we emphasize that our approach differs from these methods in its handling of classes that were unseen during training. While these methods focus on classifying unseen classes, our method is designed specifically to detect these unseen classes without performing classification, ensuring safety in real-world applications. Although there have been several studies (Bai et al., 2023; Zhu et al., 2024) to handle OOD generalization and OOD detection simultaneously, these approaches are typically limited to traditional vision models or have been evaluated with a narrow set of post-hoc functions for OOD Detection. Thus, when fine-tuning VLMs for downstream tasks, the challenge of improving the models' generalization ability to closed-set OOD data while simultaneously detecting open-set OOD classes that were unseen during fine-tuning remains largely underexplored.

**OOD detection methods** There are multiple lines of work addressing OOD detection, such as anomaly detection (Zong et al., 2018; Liang et al., 2017b), outlier detection (Bevandić et al., 2021; Saito et al., 2021), and open-set OOD recognition (Kong and Ramanan, 2021; Geng et al., 2020; Scheirer et al., 2014). These methods can be categorized into two main groups: post hoc methods (Zhu et al., 2022; Liu et al., 2020; Sun et al., 2021; Hendrycks and Gimpel, 2016; Liang et al., 2017b; Wang et al., 2022; Sun et al., 2022) and training-time regularization (Narayanaswamy et al., 2023; Bai et al., 2023; Malinin and Gales, 2018; Du et al., 2022b,a; Ming et al., 2022b). The former typically resort to post-hoc functions to recognize open-set without altering the DNN training process, like density estimation (Zhang et al., 2020), uncertainty modeling (Gal and Ghahramani, 2016), and input image reconstruction (Pidhorskyi et al., 2018; Sun et al., 2020). On the other hand, regularization-based methods aim to rectify the training process, compelling models to provide predictions with lower confidence. Recent studies (Zhou et al., 2024; Bai et al., 2024; Nie et al., 2024; Li et al., 2024a; Wang et al., 2023a; Miyai et al., 2024b; Zhang et al., 2024; Jiang et al., 2024; Ming et al., 2022a; Zhang and Zhang, 2024; Ming and Li, 2024) have explored the capability of zero-shot or few-shot OOD detection based on VLMs. More details about the CLIP-based OOD detection methods can be seen in these surveys (Miyai et al., 2024a; Li et al., 2025). However, while these studies primarily focus on handling semantic-shifted datasets, our research aims to simultaneously improve both OOD detection for semantic shifts and OOD generalization for covariate shifts, enabling more effective handling of diverse OOD datasets in real-world scenarios.

**Full-spectrum OOD detection** Recent works, such as OpenOOD v1.5 (Yang et al., 2022; Zhang et al., 2023a) and SEM (Yang et al., 2023), have taken into account both covariate shifts and semantic shifts and introduced full-spectrum OOD (FS-OOD) detection, which considers both detecting semantic shifts and being tolerant to covariate shifts. OpenOOD v1.5 extends benchmark evaluations to large-scale datasets (e.g., ImageNet) and foundation models (e.g., CLIP and DINOv2), broadening its scope to study FS-OOD detection under both types of distribution shifts. While the FS-OOD benchmark assesses OOD detection performance across diverse distribution types, it does not primarily focus on improving VLMs' classification accuracy (i.e., OOD generalization) under covariate shifts. Moreover, in practical applications, there is a strong motivation to create models that can not only detect semantically shifted OOD inputs but also generalize to covariate-shifted data. Within CLIP-based

methods, OOD detection and generalization are often discussed in separate contexts, resulting in a trade-off between detection and generalization performance (Miyai et al., 2024a). In this paper, we focus on improving VLMs' OOD generalization ability on covariate-shifted OOD data while simultaneously detecting semantic-shifted samples. To substantiate the effectiveness of the proposed method for both tasks, we conduct an empirical analysis of distinct categories of competitive methods, including the CLIP-based zero-shot OOD detection method, lightweight tuning-based OOD detection methods, and lightweight tuning-based OOD generalization methods.

## B  Proof of Theorem 3.2

We first provide the proof for property (1) in Theorem 3.2. In this paper, we achieve re-aligning vision-language modalities by disrupting the top-$c$ maximum cosine similarity to a low value. Let $\hat{y}_1 := \text{argmax}_{i \in [K]} s_i(\mathbf{x}')$ denote the index of the maximum cosine similarity for an OOD input $\mathbf{x}'$. We then reduce the maximum cosine similarity to zero according to Equation 1 and denote the difference between the original and re-aligned cosine similarity as $\tilde{s}_{\hat{y}_1}(\mathbf{x_i}) - s_{\hat{y}_1}(\mathbf{x_i}) = \mu(\mathbf{x_i})$.

Without loss of generality, we set $c = 1$, the energy score before re-alignment is denoted as $E_0(\mathbf{x}')$, which is calculated as:

$$E_0(\mathbf{x}') = -\log \sum_{i=1}^{K} e^{s_i(\mathbf{x}')/\tau} \tag{9}$$

The energy score after re-alignment is denoted as $E_1(\mathbf{x}')$, which is calculated as:

$$E_1(\mathbf{x}') = -\log[\sum_{i \neq \hat{y}_1} e^{s_i(\mathbf{x}')/\tau} + e^{\tilde{s}_{\hat{y}_1}(\mathbf{x}')/\tau}] \tag{10}$$

Given the ID class labels $\mathcal{Y}_{\text{in}}$, the newly proposed OOD score ($\Delta$Energy) is defined as:

$$S_{\Delta\text{Energy}}(\mathbf{x}'; \mathcal{Y}_{\text{in}}) = E_1(\mathbf{x}') - E_0(\mathbf{x}')$$

$$= -\log[\sum_{i \neq \hat{y}_1} e^{s_i(\mathbf{x}')/\tau} + e^{\tilde{s}_{\hat{y}_1}(\mathbf{x}')/\tau}] + \log \sum_{i=1}^{K} e^{s_i(\mathbf{x}')/\tau}$$

$$= \log \frac{\sum_{i \neq \hat{y}_1} e^{s_i(\mathbf{x}')/\tau} + e^{\tilde{s}_{\hat{y}_1}(\mathbf{x}')/\tau} + e^{s_{\hat{y}_1}(\mathbf{x}')/\tau} - e^{\tilde{s}_{\hat{y}_1}(\mathbf{x}')/\tau}}{\sum_{i \neq \hat{y}_1} e^{s_i(\mathbf{x}')/\tau} + e^{\tilde{s}_{\hat{y}_1}(\mathbf{x}')/\tau}}$$

$$= \log \left[ 1 + \frac{e^{s_{\hat{y}_1}(\mathbf{x}')/\tau} - e^{\tilde{s}_{\hat{y}_1}(\mathbf{x}')/\tau}}{\sum_{i \neq \hat{y}_1} e^{s_i(\mathbf{x}')/\tau} + e^{\tilde{s}_{\hat{y}_1}(\mathbf{x}')/\tau}} \right] \tag{11}$$

$$= \log \left[ 1 + \frac{e^{s_{\hat{y}_1}(\mathbf{x}')/\tau} - e^{(s_{\hat{y}_1}(\mathbf{x}') - \mu(\mathbf{x_i}))/\tau}}{\sum_{i \neq \hat{y}_1} e^{s_i(\mathbf{x}')/\tau} + e^{(s_{\hat{y}_1}(\mathbf{x}') - \mu(\mathbf{x_i}))/\tau}} \right]$$

$$= \log \left[ 1 + \frac{e^{\mu(\mathbf{x_i})/\tau} - 1}{\frac{\sum_{i \neq \hat{y}_1} e^{s_i(\mathbf{x}')/\tau}}{e^{(s_{\hat{y}_1}(\mathbf{x}') - \mu(\mathbf{x_i}))/\tau}} + 1} \right]$$

If we set $\tilde{s}_{\hat{y}_1}(\mathbf{x}') = 0$, i.e., $\mu(\mathbf{x_i}) = s_{\hat{y}_1}(\mathbf{x}')$, we have:

$$S_{\Delta\text{Energy}}(\mathbf{x}'; \mathcal{Y}_{\text{in}}) = E_1(\mathbf{x}') - E_0(\mathbf{x}') = \log \left[ 1 + \frac{e^{s_{\hat{y}_1}(\mathbf{x}')/\tau} - 1}{\sum_{i \neq \hat{y}_1} e^{s_i(\mathbf{x}')/\tau} + 1} \right] \tag{12}$$

Given the ID sample $\mathbf{x}_{\text{ID}}$ and OOD sample $\mathbf{x}_{\text{OOD}}$, we assume that the sum of non-maximal cosine similarities is similar for ID and OOD samples, i.e, $\sum_{i \neq \hat{y}_1} e^{s_i(\mathbf{x}_{\text{ID}})/\tau} \approx \sum_{i \neq \hat{y}_1} e^{s_i(\mathbf{x}_{\text{OOD}})/\tau}$, which is reasonable under uniform similarity distributions as discussed in prior work (Ming et al., 2022a). Since the $S_{\Delta\text{Energy}}(\mathbf{x}; \mathcal{Y}_{\text{in}})$ is a monotonically increasing function with respect to the maximum similarity $s_{\hat{y}_1}(\mathbf{x})$, the energy change for ID data is greater than that for OOD data.

We then provide the more detailed results and proof for property (2) in Theorem 3.2 as follows:

**Theorem B.1.** *[Difference Amplification between ID and OOD by* $\Delta$Energy*] Let* $S_{\Delta\text{Energy}}(\mathbf{x}, \mathcal{Y}_{in})$ *and* $S_{\Delta\text{MCM}}(\mathbf{x}, \mathcal{Y}_{in})$ *be the OOD score for any sample* $\mathbf{x}$ *by the* $\Delta$Energy *and MCM method, respectively, where:*

$$S_{\Delta\text{Energy}}(\mathbf{x}; \mathcal{Y}_{\text{in}}) = E_1(\mathbf{x}) - E_0(\mathbf{x}) = \log\left[1 + \frac{e^{s_{\hat{y}_1}(\mathbf{x})/\tau} - 1}{\sum_{i\neq\hat{y}_1} e^{s_i(\mathbf{x})/\tau} + 1}\right] \quad (13)$$

$$S_{\Delta\text{MCM}}(\mathbf{x}; \mathcal{Y}_{\text{in}}) = \frac{e^{s_{\hat{y}_1}(\mathbf{x})/\tau}}{e^{s_{\hat{y}_1}(\mathbf{x})/\tau} + \sum_{i\neq\hat{y}_1} e^{s_i(\mathbf{x})/\tau}} \quad (14)$$

*Suppose that the maximum cosine similarity for an ID sample* $\mathbf{x}_{ID}$ *is greater than that of an OOD sample* $\mathbf{x}_{OOD}$, *i.e.,* $s_{\hat{y}_1}(\mathbf{x}_{ID}) > s_{\hat{y}_1}(\mathbf{x}_{OOD})$. *We also assume that the sum of non-maximal cosine similarities is similar for ID and OOD samples, i.e,* $\sum_{i\neq\hat{y}_1} e^{s_i(\mathbf{x}_{ID})/\tau} \approx \sum_{i\neq\hat{y}_1} e^{s_i(\mathbf{x}_{OOD})/\tau}$, *which is reasonable under uniform similarity distributions as discussed in prior work (Ming et al., 2022a). Then the difference between ID and OOD under* $\Delta$Energy *exceeds that of the MCM method:*

$$d_{\Delta\text{Energy}} > d_{MCM}$$

*where* $d_{\Delta\text{Energy}} = S_{\Delta\text{Energy}}(\mathbf{x}_{ID}; \mathcal{Y}_{\text{in}}) - S_{\Delta\text{Energy}}(\mathbf{x}_{OOD}; \mathcal{Y}_{\text{in}})$, *and* $d_{MCM} = S_{\Delta\text{MCM}}(\mathbf{x}_{ID}; \mathcal{Y}_{\text{in}}) - S_{\Delta\text{MCM}}(\mathbf{x}_{OOD}; \mathcal{Y}_{\text{in}})$. *Thus, our* $\Delta$Energy *exhibits strictly stronger separability between ID and OOD distributions.*

**Proof:** Let $s = s_i(\mathbf{x})/\tau$, we simplify the $\Delta$Energy score as $S_{\Delta\text{Energy}}(\mathbf{x}, \mathcal{Y}_{\text{in}}) = \log(1 + \frac{e^s - 1}{b+1})$ and simplify the MCM score as $S_{\Delta\text{MCM}}(\mathbf{x}; \mathcal{Y}_{\text{in}}) = \frac{e^s}{e^s + b}$ where $b = \sum_{i\neq\hat{y}_1} e^{s_i(\mathbf{x})/\tau}$.

First, we analyze the gradient with respect to $s_{\hat{y}_1}$:

$$\frac{\partial S_{\Delta\text{Energy}}}{\partial s_{\hat{y}_1}} = \frac{1}{\tau} \cdot \frac{e^{s_{\hat{y}_1}/\tau}}{e^{s_{\hat{y}_1}/\tau} + b} = \frac{1}{\tau} S_{\text{MCM}} \quad (15)$$

$$\frac{\partial S_{\text{MCM}}}{\partial s_{\hat{y}_1}} = \frac{1}{\tau} \cdot \frac{e^{s_{\hat{y}_1}/\tau} b}{(e^{s_{\hat{y}_1}/\tau} + b)^2} = \frac{1}{\tau} S_{\text{MCM}}(1 - S_{\text{MCM}}) \quad (16)$$

Since $0 < S_{\text{MCM}} \leq 1$, we have $\frac{\partial S_{\Delta\text{Energy}}}{\partial s_{\hat{y}_1}} > \frac{\partial S_{\text{MCM}}}{\partial s_{\hat{y}_1}}$ and $d_{\Delta\text{Energy}} > d_{\text{MCM}}$.

Next, we consider the large-scale setting where the number of ID classes $K$ is large. Let $b \approx \sum_{i\neq\hat{y}_1} e^{s_i(\mathbf{x})/\tau}$. In this case, the denominator term $b$ becomes significantly larger than the maximum similarity term, i.e., $b \gg e^s$. We then perform Taylor Expansion for $S_{\Delta\text{Energy}}(\mathbf{x}, \mathcal{Y}_{\text{in}})$ and $S_{\Delta\text{MCM}}(\mathbf{x}, \mathcal{Y}_{\text{in}})$.

Taylor Expansion of $S_{\Delta\text{Energy}}(\mathbf{x}, \mathcal{Y}_{\text{in}})$: For $b \gg e^s$, we approximate $S_{\Delta\text{Energy}}(\mathbf{x}, \mathcal{Y}_{\text{in}})$ using $\log(1 + \epsilon) \approx \epsilon - \frac{\epsilon^2}{2}$: $S_{\Delta\text{Energy}}(\mathbf{x}, \mathcal{Y}_{\text{in}}) \approx \frac{e^s}{b} - \frac{e^{2s}}{2b^2}$.

Taylor Expansion of $S_{\Delta\text{MCM}}(\mathbf{x}, \mathcal{Y}_{\text{in}})$: Similarly, we expand $S_{\Delta\text{MCM}}(\mathbf{x}, \mathcal{Y}_{\text{in}})$: $S_{\Delta\text{MCM}}(\mathbf{x}, \mathcal{Y}_{\text{in}}) = \frac{e^s}{e^s + b} \approx \frac{e^s}{b} - \frac{e^{2s}}{b^2}$.

Then we can compute the difference between ID and OOD under the $\Delta$Energy and MCM score, respectively:

$$d_{\Delta\text{Energy}} \approx \left(\frac{e^{s_{\hat{y}_1}(\mathbf{x}_{\text{ID}})/\tau}}{b} - \frac{e^{s_{\hat{y}_1}(\mathbf{x}_{\text{OOD}})/\tau}}{b}\right) - \frac{1}{2}\left(\frac{e^{2s_{\hat{y}_1}(\mathbf{x}_{\text{ID}})/\tau}}{b^2} - \frac{e^{2s_{\hat{y}_1}(\mathbf{x}_{\text{OOD}})/\tau}}{b^2}\right) \quad (17)$$

$$d_{\text{MCM}} \approx \left(\frac{e^{s_{\hat{y}_1}(\mathbf{x}_{\text{ID}})/\tau}}{b} - \frac{e^{s_{\hat{y}_1}(\mathbf{x}_{\text{OOD}})/\tau}}{b}\right) - \left(\frac{e^{2s_{\hat{y}_1}(\mathbf{x}_{\text{ID}})/\tau}}{b^2} - \frac{e^{2s_{\hat{y}_1}(\mathbf{x}_{\text{OOD}})/\tau}}{b^2}\right) \quad (18)$$

which leads to the following property:

$$d_{\Delta\text{Energy}} - d_{\text{MCM}} \approx \frac{e^{2s_{\hat{y}_1}(\mathbf{x}_{\text{ID}})/\tau} - e^{2s_{\hat{y}_1}(\mathbf{x}_{\text{OOD}})/\tau}}{2b^2} > 0 \quad (\text{since } s_{\hat{y}_1}(\mathbf{x}_{\text{ID}}) > s_{\hat{y}_1}(\mathbf{x}_{\text{OOD}})) \quad (19)$$

In large-scale hard OOD detection scenarios where $s_{\hat{y}_1}(\mathbf{x}_{\text{ID}}) - s_{\hat{y}_1}(\mathbf{x}_{\text{OOD}}) \ll 1$ (small maximum similarity gap) and $K \gg s_{\hat{y}_1}(\mathbf{x}_{\text{ID}}) > s_{\hat{y}_1}(\mathbf{x}_{\text{OOD}})$. As demonstrated in Equation 17 and Equation 18, $\Delta$Energy's logarithmic form introduces a less aggressive decay in the higher-order term compared to MCM, preserving discriminability. For large $K$, MCM's separability diminishes quadratically ($\mathcal{O}(1/K^2)$), while $\Delta$Energy decays more slowly due to the smaller coefficient in the second-order term. Thus, $\Delta$Energy is better for large-scale OOD detection where $K$ is large and the maximum similarity gap is small ($s_{\hat{y}_1}(\mathbf{x}_{\text{ID}}) - s_{\hat{y}_1}(\mathbf{x}_{\text{OOD}}) \ll 1$).

## C   Proof of Theorem 3.3

In Theorem 3.3, we provide formal guarantees that the proposed $\Delta$Energy can provably reduce the false positive rate (FPR) compared to the widely-used VLM-based OOD detection method MCM (Ming et al., 2022a). Before the proof of Theorem 3.3, we first introduce the MCM method as follows:

For any test input $\mathbf{x}'$, we calculate the label-wise matching score based on the cosine similarity between the image feature $\mathbf{z}_{\mathbf{I}}(x')$ and the concept vector (text feature) $\mathbf{z}_{\mathbf{T}}(t_i)$: $s_i(\mathbf{x}') = \mathbf{z}_{\mathbf{I}}(\mathbf{x}') \cdot \mathbf{z}_{\mathbf{T}}(t_i)$. Note that both the image feature $\mathbf{z}_{\mathbf{I}}(x')$ and the concept vector $\mathbf{z}_{\mathbf{T}}(t_i)$ are normalized features in this paper. The maximum concept matching (MCM) score is computed as follows:

$$S_{\mathbf{MCM}}(\mathbf{x}'; \mathcal{Y}_{\text{in}}) = \max_{i} \frac{e^{s_i(\mathbf{x}')/\tau}}{\sum_{j=1}^{K} e^{s_j(\mathbf{x}')/\tau}},$$

Under the Assumption C.1, Ming et.al (Ming et al., 2022a) have provided the formal guarantees for MCM that using softmax can provably reduce the false positive rate (FPR) compared to that without softmax, as illustrated in Theorem C.2.

**Assumption C.1.  [(Ming et al., 2022a)]** Let $z := \mathbb{1}\{y \in \mathcal{Y}_{\text{in}}\}$.   $Q_{\mathbf{x}}$ denotes the out-of-distribution $\mathbb{P}_{\mathbf{x}|z=0}$ (marginal distribution of x conditioned on $z = 0$). Assume $\exists \delta > 0$ such that

$$Q_{\mathbf{x}} \left( \frac{1}{K-1} \sum_{i \neq \hat{y}_1} [s_{\hat{y}_2}(\mathbf{x}) - s_i(\mathbf{x})] \leqslant \delta \right) = 1,$$

where $\hat{y}_1 := \operatorname{argmax}_{i \in [K]} s_i(\mathbf{x})$ and $\hat{y}_2 := \operatorname{argmax}_{i \neq \hat{y}_1, i \in [K]} s_i(\mathbf{x})$ denote the indices of the largest and second-largest cosine similarities for an OOD input $\mathbf{x}$.

**Theorem C.2.  [(Ming et al., 2022a)]** *Given a task with ID label set $\mathcal{Y}_{\text{in}} = \{y_1, y_2, ..., y_K\}$ and a pre-trained VLM. If $Q_{\mathbf{x}}$ satisfies Assumption C.1, then there exists a constant $T = \frac{\lambda(K-1)(\lambda^{wo}+\delta-s_{\hat{y}2})}{K\lambda-1}$ such that for any temperature $\tau > T$, we have*

$$\text{FPR}(\tau, \lambda) \leq \text{FPR}^{\text{wo}}(\lambda^{\text{wo}}),$$

*where $\text{FPR}(\tau, \lambda)$ is the false positive rate based on softmax scaling with temperature $\tau$ and detection threshold $\lambda$; $\text{FPR}^{\text{wo}}(\lambda^{\text{wo}})$ is the false positive rate without softmax scaling based on threshold $\lambda^{\text{wo}}$.*

Now we present the proof of Theorem 3.3 as follows:

**Proof:** The newly proposed OOD score ($\Delta$Energy) is defined as:

$$
S_{\Delta\text{Energy}}(\mathbf{x}'; \mathcal{Y}_{\text{in}}) = E_1 - E_0
$$

$$
= -\log\Big[\sum_{i\neq\hat{y}_1} e^{s_i(\mathbf{x}')/\tau} + e^{\tilde{s}_{\hat{y}_1}(\mathbf{x}')/\tau}\Big] + \log\sum_{i=1}^{K} e^{s_i(\mathbf{x}')/\tau}
$$

$$
= \log \frac{\sum_{i\neq\hat{y}_1} e^{s_i(\mathbf{x}')/\tau} + e^{\tilde{s}_{\hat{y}_1}(\mathbf{x}')/\tau} + e^{s_{\hat{y}_1}(\mathbf{x}')/\tau} - e^{\tilde{s}_{\hat{y}_1}(\mathbf{x}')/\tau}}{\sum_{i\neq\hat{y}_1} e^{s_i(\mathbf{x}')/\tau} + e^{\tilde{s}_{\hat{y}_1}(\mathbf{x}')/\tau}}
$$

$$
= \log\left[1 + \frac{e^{s_{\hat{y}_1}(\mathbf{x}')/\tau} - e^{\tilde{s}_{\hat{y}_1}(\mathbf{x}')/\tau}}{\sum_{i\neq\hat{y}_1} e^{s_i(\mathbf{x}')/\tau} + e^{\tilde{s}_{\hat{y}_1}(\mathbf{x}')/\tau}}\right]
$$

$$
\leq \frac{e^{s_{\hat{y}_1}(\mathbf{x}')/\tau} - e^{\tilde{s}_{\hat{y}_1}(\mathbf{x}')/\tau}}{\sum_{i\neq\hat{y}_1} e^{s_i(\mathbf{x}')/\tau} + e^{\tilde{s}_{\hat{y}_1}(\mathbf{x}')/\tau}} = \frac{e^{s_{\hat{y}_1}(\mathbf{x}')/\tau} - e^{\tilde{s}_{\hat{y}_1}(\mathbf{x}')/\tau}}{\sum_{i=1}^{K} e^{s_i(\mathbf{x}')/\tau} + e^{\tilde{s}_{\hat{y}_1}(\mathbf{x}')/\tau} - e^{s_{\hat{y}_1}(\mathbf{x}')/\tau}}
$$

$$
\leq \frac{e^{s_{\hat{y}_1}(\mathbf{x}')/\tau}}{\sum_{i=1}^{K} e^{s_i(\mathbf{x}')/\tau} + 2e^{\tilde{s}_{\hat{y}_1}(\mathbf{x}')/\tau} - e^{s_{\hat{y}_1}(\mathbf{x}')/\tau}}
$$

$$(20)$$

When $c \in \{2, \cdots, K\}$, without loss of generality, we take $c = 2$ as example and we have:

$$
S_{\Delta\text{Energy}}(\mathbf{x}'; \mathcal{Y}_{\text{in}}) = E_1 - E_0
$$

$$
= \frac{1}{2}\left[-\log\Big[\sum_{i\neq\hat{y}_1} e^{s_i(\mathbf{x}')/\tau} + e^{\tilde{s}_{\hat{y}_1}(\mathbf{x}')/\tau}\Big] + \log\sum_{i=1}^{K} e^{s_i(\mathbf{x}')/\tau}\right]
$$

$$
+ \frac{1}{2}\left[-\log\Big[\sum_{i\neq\hat{y}_2} e^{s_i(\mathbf{x}')/\tau} + e^{\tilde{s}_{\hat{y}_2}(\mathbf{x}')/\tau}\Big] + \log\sum_{i=1}^{K} e^{s_i(\mathbf{x}')/\tau}\right]
$$

$$(21)$$

$$
\leq \frac{1}{2}\frac{e^{s_{\hat{y}_1}(\mathbf{x}')/\tau}}{\sum_{i=1}^{K} e^{s_i(\mathbf{x}')/\tau} + 2e^{\tilde{s}_{\hat{y}_1}(\mathbf{x}')/\tau} - e^{s_{\hat{y}_1}(\mathbf{x}')/\tau}}
$$

$$
+ \frac{1}{2}\frac{e^{s_{\hat{y}_2}(\mathbf{x}')/\tau}}{\sum_{i=1}^{K} e^{s_i(\mathbf{x}')/\tau} + 2e^{\tilde{s}_{\hat{y}_2}(\mathbf{x}')/\tau} - e^{s_{\hat{y}_2}(\mathbf{x}')/\tau}}
$$

For $c = 1$, if $2e^{\tilde{s}_{\hat{y}_1}(\mathbf{x}')/\tau} - e^{s_{\hat{y}_1}(\mathbf{x}')/\tau} \geq 0$, i.e., $s_{\hat{y}_1}(\mathbf{x}') - \tilde{s}_{\hat{y}_1}(\mathbf{x}') \leq \tau\ln 2$, we have

$$
S_{\Delta\text{Energy}}(\mathbf{x}'; \mathcal{Y}_{\text{in}}) \leq \frac{e^{s_{\hat{y}_1}(\mathbf{x}')/\tau}}{\sum_{i=1}^{K} e^{s_i(\mathbf{x}')/\tau} + 2e^{\tilde{s}_{\hat{y}_1}(\mathbf{x}')/\tau} - e^{s_{\hat{y}_1}(\mathbf{x}')/\tau}} \leq \frac{e^{s_{\hat{y}_1}(\mathbf{x}')/\tau}}{\sum_{i=1}^{K} e^{s_i(\mathbf{x}')/\tau}} = S_{\text{MCM}}(\mathbf{x}'; \mathcal{Y}_{\text{in}})
$$

For $c = 2$, we have $s_{\hat{y}_2}(\mathbf{x}') \leq s_{\hat{y}_1}(\mathbf{x}')$, if $2e^{\tilde{s}_{\hat{y}_2}(\mathbf{x}')/\tau} - e^{s_{\hat{y}_2}(\mathbf{x}')/\tau} \geq 0$, i.e., $s_{\hat{y}_2}(\mathbf{x}') - \tilde{s}_{\hat{y}_2}(\mathbf{x}') \leq \tau\ln 2$, we have

$$
\frac{e^{s_{\hat{y}_2}(\mathbf{x}')/\tau}}{\sum_{i=1}^{K} e^{s_i(\mathbf{x}')/\tau} + 2e^{\tilde{s}_{\hat{y}_2}(\mathbf{x}')/\tau} - e^{s_{\hat{y}_2}(\mathbf{x}')/\tau}} \leq \frac{e^{s_{\hat{y}_1}(\mathbf{x}')/\tau}}{\sum_{i=1}^{K} e^{s_i(\mathbf{x}')/\tau} + 2e^{\tilde{s}_{\hat{y}_2}(\mathbf{x}')/\tau} - e^{s_{\hat{y}_2}(\mathbf{x}')/\tau}}
$$

$$(22)$$

$$
\leq \frac{e^{s_{\hat{y}_1}(\mathbf{x}')/\tau}}{\sum_{i=1}^{K} e^{s_i(\mathbf{x}')/\tau}} = S_{\text{MCM}}(\mathbf{x}'; \mathcal{Y}_{\text{in}})
$$

which leads to

$$
S_{\Delta\text{Energy}}(\mathbf{x}'; \mathcal{Y}_{\text{in}}) \leq S_{\text{MCM}}(\mathbf{x}'; \mathcal{Y}_{\text{in}})
$$

Let the OOD detection functions be represented by:

$$
G(\mathbf{x}'; \mathcal{Y}_{\text{in}}) = \begin{cases} 1 & S_{\Delta\text{Energy}}(\mathbf{x}'; \mathcal{Y}_{\text{in}}) \geq \lambda \\ 0 & S_{\Delta\text{Energy}}(\mathbf{x}'; \mathcal{Y}_{\text{in}}) < \lambda \end{cases},
$$

$$(23)$$

then we have

$$
\begin{aligned}
\text{FPR}^{\Delta\text{Energy}}(\tau, \lambda) &= \mathbb{P}\left(G(\mathbf{x}'; \mathcal{Y}_{\text{in}}) = 1 \mid z = 0\right) \\
&= Q_{\mathbf{x}'}\left(G(\mathbf{x}'; \mathcal{Y}_{\text{in}}) = 1\right) \\
&= Q_{\mathbf{x}'}\left(S_{\Delta\text{Energy}}(\mathbf{x}'; \mathcal{Y}_{\text{in}}) > \lambda\right) \\
&\leq Q_{\mathbf{x}'}\left(S_{\text{MCM}}(\mathbf{x}'; \mathcal{Y}_{\text{in}}) > \lambda\right) = \text{FPR}^{\text{MCM}}(\tau, \lambda)
\end{aligned}
\tag{24}
$$

Thus, we complete the proof.

## D  Proof of Theorem 3.4

Proof: To further enlarge the energy change between the masked VLM and unmasked VLM for closed-set classes, we propose to minimize the following loss:

$$
\mathcal{L}_{\Delta E} = \frac{1}{N} \sum_{i=1}^{N} E_2(\mathbf{x_i}) - E_0(\mathbf{x_i})
\tag{25}
$$

where $E_2(\mathbf{x_i})$ is the energy score for $\mathbf{x_i}$ after masking on the image feature, which is formally calculated as:

$$
E_2(\mathbf{x_i}) = -\log \sum_{j=1}^{K} e^{s'_j(\mathbf{x_i})}
$$

$$
s'_j(\mathbf{x_i}) = (\mathbf{z_I}(\mathbf{x_i}) \odot \mathbf{m}'(\mathbf{x_i})) \cdot \mathbf{z_T}(t_j)
$$

where $\mathbf{m}'(\mathbf{x_i})$ is the mask that retains the top $p$-proportion elements in $\mathbf{z_I}(\mathbf{x_i}) \odot \mathbf{h_1}(\mathbf{x_i})$

Now we prove that $-\mathcal{L}_{\Delta E}$ is the lower bound of $\sum_{i=1}^{N} \Delta\text{Energy}(\mathbf{x_i})$. Here, we represent the optimization term for $\mathbf{x_i}$ as: $\mathcal{L}_{\Delta E}(\mathbf{x_i}) := E_2(\mathbf{x_i}) - E_0(\mathbf{x_i})$. Then the relationship between $\Delta\text{Energy}(\mathbf{x_i})$ and $\mathcal{L}_{\Delta E}(\mathbf{x_i})$ can be formulated as:

$$
\begin{aligned}
e^{\Delta\text{Energy}(\mathbf{x_i})} - e^{-\mathcal{L}_{\Delta E}(\mathbf{x_i})} &= \frac{\sum_{j=1}^{K} e^{s_j(\mathbf{x_i})/\tau}}{\sum_{j \neq \hat{y}_1} e^{s_j(\mathbf{x_i})/\tau} + e^{\tilde{s}_{\hat{y}_1}(\mathbf{x_i})/\tau}} - \frac{\sum_{j \neq \hat{y}_1} e^{s'_j(\mathbf{x_i})/\tau} + e^{s'_{\hat{y}_1}(\mathbf{x_i})/\tau}}{\sum_{j=1}^{K} e^{s_j(\mathbf{x_i})/\tau}} \\
&= \frac{\sum_{j=1}^{K} e^{s_j(\mathbf{x_i})/\tau}}{\sum_{j \neq \hat{y}_1} e^{s_j(\mathbf{x_i})/\tau} + e^{\tilde{s}_{\hat{y}_1}(\mathbf{x_i})/\tau}} - \frac{\sum_{j \neq \hat{y}_1} e^{s'_j(\mathbf{x_i})/\tau} + e^{s'_{\hat{y}_1}(\mathbf{x_i})/\tau}}{\sum_{j=1}^{K} e^{s_j(\mathbf{x_i})/\tau}} \\
&= \frac{e^{s_{\hat{y}_1}(\mathbf{x_i})/\tau} - e^{\tilde{s}_{\hat{y}_1}(\mathbf{x_i})/\tau}}{\sum_{j \neq \hat{y}_1} e^{s_j(\mathbf{x_i})/\tau} + e^{\tilde{s}_{\hat{y}_1}(\mathbf{x_i})/\tau}} - \frac{\sum_{j=1}^{K}[e^{s'_j(\mathbf{x_i})/\tau} - e^{s_j(\mathbf{x_i})/\tau}]}{\sum_{j=1}^{K} e^{s_j(\mathbf{x_i})/\tau}} \\
&\geq \frac{e^{s_{\hat{y}_1}(\mathbf{x_i})/\tau} - e^{\tilde{s}_{\hat{y}_1}(\mathbf{x_i})/\tau}}{\sum_{j \neq \hat{y}_1} e^{s_j(\mathbf{x_i})/\tau} + e^{s_{\hat{y}_1}(\mathbf{x_i})/\tau}} - \frac{\sum_{j=1}^{K}[e^{s'_j(\mathbf{x_i})/\tau} - e^{s_j(\mathbf{x_i})/\tau}]}{\sum_{j=1}^{K} e^{s_j(\mathbf{x_i})/\tau}}
\end{aligned}
\tag{26}
$$

where the inequality in the last line follows $\tilde{s}_{\hat{y}_1}(\mathbf{x_i}) \leq s_{\hat{y}_1}(\mathbf{x_i})$.

Under the condition that $e^{s_{\hat{y}_1}(\mathbf{x_i})/\tau} - e^{\tilde{s}_{\hat{y}_1}(\mathbf{x_i})/\tau} \geq (e^{\varepsilon_E} - 1)\sum_{j=1}^{K} e^{s'_j(\mathbf{x_i})/\tau} = (e^{\varepsilon_E} - 1)e^{-E_2(\mathbf{x_i})}$ and that $\mathcal{L}_{\Delta E} \leq \varepsilon_E$, it is straightforward to derive the following inequality:

$$
e^{s_{\hat{y}_1}(\mathbf{x_i})/\tau} - e^{\tilde{s}_{\hat{y}_1}(\mathbf{x_i})/\tau} \geq \sum_{j=1}^{K}[e^{s'_j(\mathbf{x_i})/\tau} - e^{s_j(\mathbf{x_i})/\tau}]
$$

Thus we have $\Delta\text{Energy}(\mathbf{x_i}) \geq -\mathcal{L}_{\Delta E}(\mathbf{x_i})$. Thus complete the proof.

## E  Proof of Theorem 3.5

**Proof:** In the prompt-tuning framework of our proposed EBM, only $n$ context vectors are learnable, and we denote the learnable context vectors as $\theta = [\theta_1, \cdots, \theta_n]$. The $\mathcal{L}_{\Delta E}$ can be represented by:

$$
\mathcal{L}_{\Delta E} = \frac{1}{N} \sum_{i=1}^{N} \left( \log \sum_{j=1}^{K} \exp \langle \mathbf{z_I}(\mathbf{x_i}), \mathbf{z_T}(t_j; \theta) \rangle - \log \sum_{j=1}^{K} \exp \langle \mathbf{m_I}(\mathbf{x_i}), \mathbf{z_T}(t_j; \theta) \rangle \right)
\tag{27}
$$

where $\mathbf{m_I}(\mathbf{x_i}) = \mathbf{z_I}(\mathbf{x_i}) \odot \mathbf{m'}(\mathbf{x_i})$ is the masked image feature. Now we expand $\nabla_\theta \mathcal{L}_{\Delta E}$ as follows:

$$\nabla_\theta \mathcal{L}_{\Delta E} = \frac{1}{N} \sum_{i=1}^{N} \left[ \frac{\nabla_\theta \sum_{j=1}^{K} \exp \langle \mathbf{z_I}(\mathbf{x_i}), \mathbf{z_T}(t_j; \theta) \rangle}{\sum_{j=1}^{K} \exp \langle \mathbf{z_I}(\mathbf{x_i}), \mathbf{z_T}(t_j; \theta) \rangle} - \frac{\nabla_\theta \sum_{j=1}^{K} \exp \langle \mathbf{m_I}(\mathbf{x_i}), \mathbf{z_T}(t_j; \theta) \rangle}{\sum_{j=1}^{K} \exp \langle \mathbf{m_I}(\mathbf{x_i}), \mathbf{z_T}(t_j; \theta) \rangle} \right] \tag{28}$$

We denote

$$\mathbf{a_0} = \frac{1}{N} \sum_{i=1}^{N} \left( \log \sum_{j=1}^{K} \exp \langle \mathbf{z_I}(\mathbf{x_i}), \mathbf{z_T}(t_j; \theta) \rangle \right)$$

$$\mathbf{a_1} = \frac{1}{N} \sum_{i=1}^{N} \left( \log \sum_{j=1}^{K} \exp \langle \mathbf{m_I}(\mathbf{x_i}), \mathbf{z_T}(t_j; \theta) \rangle \right)$$

Then we have $\nabla_\theta \mathcal{L}_{\Delta E} = \nabla_\theta \mathbf{a_0} - \nabla_\theta \mathbf{a_1}$. The local optimization of $\mathcal{L}_{\Delta E}$ lead to $\nabla_\theta \mathbf{a_0} = \nabla_\theta \mathbf{a_1}$.

Let $S^{(i)}$ represent the cosine similarity between the image feature $\mathbf{z_I}(\mathbf{x_i})$ and the text feature corresponding to its ground-truth label. The empirical classification loss, $\mathcal{E}_\mathcal{D}(\theta)$, can be calculated as:

$$\begin{aligned} \mathcal{E}_\mathcal{D}(\theta) &= -\frac{1}{N} \sum_{i=1}^{N} \log \frac{\exp S^{(i)}}{\sum_{j=1}^{K} \exp \langle \mathbf{z_I}(\mathbf{x_i}), \mathbf{z_T}(t_j; \theta) \rangle} \\ &= \frac{1}{N} \sum_{i=1}^{N} \left[ \log \sum_{j=1}^{K} \exp \langle \mathbf{z_I}(\mathbf{x_i}), \mathbf{z_T}(t_j; \theta) \rangle - S^{(i)} \right] \end{aligned} \tag{29}$$

Accordingly, the gradient vector of empirical risk $\widehat{\mathcal{E}}_\mathcal{D}(\theta)$ with respect to parameter $\theta$ is represented as:

$$\widehat{\mathbf{G}}_\mathcal{D}(\theta) = \nabla_\theta \widehat{\mathcal{E}}_\mathcal{D}(\theta) = \frac{1}{N} \sum_{i=1}^{N} \left[ \frac{\nabla_\theta \sum_{j=1}^{K} \exp \langle \mathbf{z_I}(\mathbf{x_i}), \mathbf{z_T}(t_j; \theta) \rangle}{\sum_{j=1}^{K} \exp \langle \mathbf{z_I}(\mathbf{x_i}), \mathbf{z_T}(t_j; \theta) \rangle} - \nabla_\theta S^{(i)} \right] = -\mathbf{a} - \frac{1}{N} \sum_{i=1}^{N} \nabla_\theta S^{(i)} \tag{30}$$

And the Hessian matrix of empirical risk with respect to parameter $\theta$ is calculated as:

$$\widehat{\mathbf{H}}_\mathcal{D}(\theta) = \nabla_\theta^2 \widehat{\mathcal{E}}_\mathcal{D}(\theta) = -\nabla_\theta \mathbf{a} - \frac{1}{N} \sum_{i=1}^{N} \nabla_\theta^2 S^{(i)} \tag{31}$$

The local optimum solution of Equation 4, i.e., $\nabla_\theta \mathcal{L}_{\Delta E} = \mathbf{0}$, gives the following equation:

$$\widehat{\mathbf{H}}_\mathcal{S}(\theta) - \widehat{\mathbf{H}}_{\mathcal{S'}}(\theta) = -\frac{1}{N} \sum_{i=1}^{N} (\nabla_\theta^2 S^{(i)} - \nabla_\theta^2 S_m^{(i)}) = -\frac{1}{N} \sum_{i=1}^{N} \nabla_\theta^2 [(\mathbf{z_I}(\mathbf{x_i}) - \mathbf{m_I^{(i)}}) \cdot \mathbf{z_T}(\mathbf{x_i})] \tag{32}$$

Finally, we can conclude that the local optimum solution of Equation 4 leads to the following property:

$$|\theta^\top (\widehat{\mathbf{H}}_\mathcal{S}(\theta) - \widehat{\mathbf{H}}_{\mathcal{S'}}(\theta))\theta| = |\theta^\top \frac{1}{N} \sum_{i=1}^{N} \nabla_\theta^2 [(\mathbf{z_I}(\mathbf{x_i}) - \mathbf{m_I^{(i)}}) \cdot \mathbf{z_T}(\mathbf{x_i})\theta]| \le O(\varepsilon) \tag{33}$$

## F  Proof of Proposition 3.6

**Proof:** Let $\theta^*$ be the local minimum across all domains, i.e., $\nabla_\theta \widehat{\mathcal{E}}_\mathcal{D}(\theta^*) = \mathbf{0}$, and $\mathcal{D} = \{\mathcal{S}, \mathcal{T}\}$. By Taylor expansion, the OOD generalization gap between source domain ($\mathcal{S}$) and target domain ($\mathcal{T}$) is

Table 5: **Conventional OOD detection Results:** OOD detection performance for ImageNet-1k as ID. In the table, we extend our $\Delta$Energy method to the zero-shot method CSP, leveraging the informative information from extra OOD labels as detailed in Equation 36.

| | Texture | | iNaturalist | | Places | | SUN | | Avg | |
|---|---|---|---|---|---|---|---|---|---|---|
| Method | AUC↑ | FPR95↓ | AUC↑ | FPR95↓ | AUC↑ | FPR95↓ | AUC↑ | FPR95↓ | AUC↑ | FPR95↓ |
| CLIP-based post-hoc methods | | | | | | | | | | |
| MSP | 74.84 | 73.66 | 77.74 | 74.57 | 72.18 | 79.12 | 73.97 | 76.95 | 74.98 | 76.22 |
| MaxLogit | 88.63 | 48.72 | 88.03 | 60.88 | 87.45 | 55.54 | 91.16 | 44.83 | 88.82 | 52.49 |
| Energy | 88.22 | 50.39 | 87.18 | 64.98 | 87.33 | 57.40 | 91.17 | 46.42 | 88.48 | 54.80 |
| ReAct | 88.13 | 49.88 | 86.87 | 65.57 | 87.42 | 56.85 | 91.04 | 46.17 | 88.37 | 54.62 |
| ODIN | 87.85 | 51.67 | 94.65 | 30.22 | 85.54 | 55.06 | 87.17 | 54.04 | 88.80 | 47.75 |
| Tuning-based methods | | | | | | | | | | |
| NegPrompt | 91.60 | 35.21 | 98.73 | 6.32 | 93.34 | 27.60 | 95.55 | 22.89 | 94.81 | 23.01 |
| ID-Like | 94.32 | 25.27 | 98.19 | 8.98 | 91.15 | 41.74 | 91.64 | 42.03 | 93.83 | 29.51 |
| LoCoOp | 90.19 | 42.28 | 96.86 | 16.05 | 91.98 | 32.87 | 95.07 | 23.44 | 93.52 | 28.66 |
| LSN+CoOp | 89.52 | 31.57 | 95.47 | 23.48 | 90.87 | 36.43 | 93.45 | 29.84 | 92.33 | 31.97 |
| LSN+CoCoOp | 90.42 | 38.54 | 95.83 | 21.56 | 91.25 | 34.48 | 94.35 | 26.32 | 92.96 | 30.22 |
| GalLoP | 90.40 | 38.40 | 97.10 | 13.70 | 91.30 | 32.50 | 94.00 | 24.90 | 93.20 | 27.30 |
| Zero-shot methods | | | | | | | | | | |
| MCM | 86.11 | 57.77 | 94.61 | 30.91 | 89.77 | 44.69 | 92.57 | 34.59 | 90.76 | 42.74 |
| CLIPN | 90.93 | 40.83 | 95.27 | 23.94 | 92.28 | 33.45 | 93.92 | 26.17 | 93.10 | 31.10 |
| NegLabel | 90.22 | 43.56 | 99.49 | 1.91 | 91.64 | 35.59 | 95.49 | 20.53 | 94.21 | 25.40 |
| CSP | 93.86 | 25.52 | 99.60 | 1.54 | **92.90** | 29.32 | **96.66** | **13.66** | 95.76 | 17.51 |
| CSP+$\Delta$Energy (Ours) | **94.33** | **21.44** | **99.72** | **0.82** | 92.66 | **28.87** | 96.60 | 13.75 | **95.83** | **16.22** |

Table 6: **Hard OOD detection Results #1:** OOD detection measured by AUROC and FPR95 over 4 different splits of ImageNet-1k. Details of the 4 splits are in Table 8.

| | Split-1 | | Split-2 | | Split-3 | | Split-4 | | Avg | |
|---|---|---|---|---|---|---|---|---|---|---|
| Method | AUC↑ | FPR95↓ | AUC↑ | FPR95↓ | AUC↑ | FPR95↓ | AUC↑ | FPR95↓ | AUC↑ | FPR95↓ |
| MCM | 97.93 | 9.17 | 88.10 | 56.40 | 90.34 | 33.05 | 98.72 | 4.73 | 93.77 | 25.83 |
| CLIPN | 99.38 | 2.07 | 97.77 | 10.55 | 90.03 | 36.85 | 98.83 | 4.68 | 96.50 | 13.53 |
| MSP | 77.85 | 63.60 | 68.73 | 83.63 | 79.10 | 70.55 | 82.40 | 65.52 | 77.02 | 70.83 |
| MaxLogit | 99.87 | 0.49 | 98.06 | 8.69 | 90.96 | 34.34 | 99.35 | **2.66** | 97.06 | 11.55 |
| Energy | 99.88 | 0.46 | 98.18 | 8.40 | 90.65 | 35.02 | 99.36 | 2.83 | 97.02 | 11.68 |
| ReAct | 99.34 | 0.72 | 97.91 | 9.33 | 90.72 | 35.65 | 99.12 | 2.94 | 96.77 | 12.16 |
| ODIN | 98.78 | 1.12 | 98.23 | 8.18 | 89.92 | 37.20 | 98.76 | 13.20 | 96.42 | 14.92 |
| $\Delta$Energy (Ours) | **99.93** | **0.37** | **99.00** | **5.16** | **91.14** | **30.09** | **99.40** | 2.83 | **97.37** | **9.61** |

upper bounded as shown in the following equation:

$$
\begin{aligned}
&\max_{\{\theta:|\widehat{\mathcal{E}}_{\mathcal{S}}(\theta)-\widehat{\mathcal{E}}_{\mathcal{S}}(\theta^*)|\leq\epsilon\}} |\widehat{\mathcal{E}}_{\mathcal{T}}(\theta) - \widehat{\mathcal{E}}_{\mathcal{S}}(\theta^*)| \\
&\approx \max_{\{\theta:\frac{1}{2}|\theta^\top\widehat{\mathbf{H}}_{\mathcal{S}}(\theta^*)\theta|\leq\epsilon\}} \left|\widehat{\mathcal{E}}_{\mathcal{T}}(\theta^*) + \frac{1}{2}\theta^\top\widehat{\mathbf{H}}_{\mathcal{T}}(\theta^*)\theta - \widehat{\mathcal{E}}_{\mathcal{S}}(\theta^*)\right| \\
&\lesssim |\widehat{\mathcal{E}}_{\mathcal{T}}(\theta^*) - \widehat{\mathcal{E}}_{\mathcal{S}}(\theta^*)| + \max_{\{\theta:\frac{1}{2}|\theta^\top\widehat{\mathbf{H}}_{\mathcal{S}}(\theta^*)\theta|\leq\epsilon\}} \frac{1}{2}\left|\theta^\top\widehat{\mathbf{H}}_{\mathcal{T}}(\theta^*)\theta\right| \qquad (34) \\
&\lesssim |\widehat{\mathcal{E}}_{\mathcal{T}}(\theta^*) - \widehat{\mathcal{E}}_{\mathcal{S}}(\theta^*)| + \max_{\{\theta:\frac{1}{2}|\theta^\top\widehat{\mathbf{H}}_{\mathcal{S}}(\theta^*)\theta|\leq\epsilon\}} \frac{1}{2}\left|\theta^\top[\widehat{\mathbf{H}}_{\mathcal{T}}(\theta^*) - \widehat{\mathbf{H}}_{\mathcal{S}}(\theta^*) + \widehat{\mathbf{H}}_{\mathcal{S}}(\theta^*)]\theta\right| \\
&\lesssim |\widehat{\mathcal{E}}_{\mathcal{T}}(\theta^*) - \widehat{\mathcal{E}}_{\mathcal{S}}(\theta^*)| + \max \frac{1}{2}|\theta^\top[\widehat{\mathbf{H}}_{\mathcal{T}}(\theta^*) - \widehat{\mathbf{H}}_{\mathcal{S}}(\theta^*)]\theta| + \epsilon
\end{aligned}
$$

For each image feature $\mathbf{z_I}$ from the source domain, the image features $\tilde{\mathbf{z}}_{\mathbf{I}}$ from the target domain, which share the same label with $\mathbf{z_I}$, is assumed to satisfy: $||\mathbf{z_I} - \tilde{\mathbf{z}}_{\mathbf{I}}||_2 \leq \varepsilon_1$. Since we optimize the EBM loss based on the unmasked image features and masked image features, we have the following approximation:

$$
\left|\theta^\top[\widehat{\mathbf{H}}_{\mathcal{T}}(\theta^*) - \widehat{\mathbf{H}}_{\mathcal{S}}(\theta^*)](\theta)\theta\right| \leq O(\varepsilon_1) \qquad (35)
$$

## G  More Experiment Results

**More experiment details** We present experiment details for the baseline models as follows:

Table 7: **Hard OOD detection Results #2.** Comparison with state-of-the-art zero-shot methods on hard OOD detection datasets. In the table, OOD detection is measured by AUROC and FPR95 over 6 hard OOD detection datasets. Details of those datasets can be seen in prior researches (Chen et al., 2024; Ming et al., 2022a).

| ID datasets
OOD datasets | ImageNet-10
ImageNet-20 | | ImageNet-20
ImageNet-10 | | ImageNet-10
ImageNet-100 | | ImageNet-100
ImageNet-10 | | ImageNet-1k
ImageNet-O | | WaterBirds
Placesbg | |
|---|---|---|---|---|---|---|---|---|---|---|---|---|
| Method | AUROC | FPR95 | AUROC | FPR95 | AUROC | FPR95 | AUROC | FPR95 | AUROC | FPR95 | AUROC | FPR95 |
| MCM | 98.60 | 6.00 | 98.09 | 13.04 | 99.39 | 2.50 | 87.20 | 60.00 | 78.59 | 64.27 | 87.45 | 33.62 |
| NegLabel | 98.80 | 5.00 | 98.04 | 11.60 | 99.37 | 2.50 | 87.93 | 49.40 | 85.78 | 56.65 | 87.99 | 29.16 |
| CSP | 99.02 | 3.30 | 98.79 | 3.40 | 99.33 | 2.22 | 89.59 | **42.40** | 88.08 | 51.50 | 92.88 | 12.07 |
| $\Delta$Energy (Ours) | **99.11** | **2.80** | **99.01** | **3.20** | **99.40** | **1.80** | **91.05** | 44.80 | **90.49** | **41.25** | **93.45** | **11.00** |

Table 8: The 4 ImageNet-1k splits for hard OOD detection following the prior work (Li et al., 2024a). Given are the numbers of classes : training / test samples.

| | ID | OOD |
|---|---|---|
| **Split-1** | All dog classes
116: 1856 / 5800 | Non-animal classes
166: — / 8300 |
| **Split-2** | Half of hunting dog classes
30: 480 / 1500 | Other 4-legged animal classes
55: — / 2750 |
| **Split-3** | Mix of common classes
151: 2416 / 7550 | Mix of common classes
164: — / 8200 |
| **Split-4** | First 100 classes
100: 1600 / 5000 | Remaining 900 classes
900: — / 45000 |

---

**Algorithm 1** Algorithm of the proposed EBM method

---

1: **Input:** ID data $\{\mathbf{x_i}, \mathbf{y_i}\}$ $(i \in 1, \cdots, N)$, ID class names of the $K$-way classification, masking proportion $p$, text prompts $\{t_1, t_2, \cdot, t_K\}$, hyperparameter $\lambda_0$, and maximum epoch $T$.
2: **for** $t = 1$ **to** $T$ **do**
3:     Calculate the ID image features $\mathbf{z_I}(\mathbf{x_i})$ and fine-tuned text features $\mathbf{z_T}(t_j; \theta)$ when prompt-tuning the VLM;
4:     Compute the cosine similarity between image features $\mathbf{z_I}(\mathbf{x_i})$ and text features $\mathbf{z_T}(t_j; \theta)$ and denote the text feature with the top-1 similarity as $\mathbf{h_1}(\mathbf{x_i}; \theta)$ ;
5:     Compute the element-wise product $\mathbf{z_P}(\mathbf{x_i}) := \mathbf{z_I}(\mathbf{x_i}) \odot \mathbf{h_1}(\mathbf{x_i}; \theta)$ and generate the mask, denoted as $\mathbf{m}'(\mathbf{x_i})$, which retains the top $p$-proportion elements in $\mathbf{z_P}(\mathbf{x_i})$;
6:     Perform masking on the image feature and represent the masked image feature as $\mathbf{z_I}(\mathbf{x_i}) \odot \mathbf{m}'(\mathbf{x_i})$;
7:     Gradient update under the proposed loss as illustrated in Equation 8;
8: **end for**
9: **Output:** Learnable content vectors $\theta$.

---

For zero-shot OOD detection methods such as CSP and NegLabel, all hyperparameters and OOD score calculation procedures are directly adopted from their respective papers (Chen et al., 2024; Jiang et al., 2024) without modification.

For the tuning-based methods, based on the code of CoOp (Zhou et al., 2021), we train models with SGD optimizer with a learning rate of $2e - 2$. The batch size is set to 32 for all tuning-based experiments. For the specific hyperparameter for each method, we follow the setting of the original paper.

For the tuning-based methods for improving performances on closed-set data, such as CoOp (Zhou et al., 2021), CoCoOp (Zhou et al., 2022), DPLCLIP (Zhang et al., 2021b), and Bayes-CAL (Zhu et al., 2023b), we use random initialization for context vectors and set the number of context tokens as 16, set the class token position (CTP) as "end", and set the class-specific context (CSC) as "False". This configuration has shown the best average performance according to CoOp's paper. For the DPLCLIP (Zhang et al., 2021b) method, we set the additional hyperparameters of DPLCLIP (Zhang et al., 2021b) as: "mlp_depth=3", "mlp_width=512", and "mlp_dropout=0.1". For the CLIP-Adapter (Gao et al., 2023) method, we adopt image adapter only with the residual ratio of 0.5 for Setup-I and

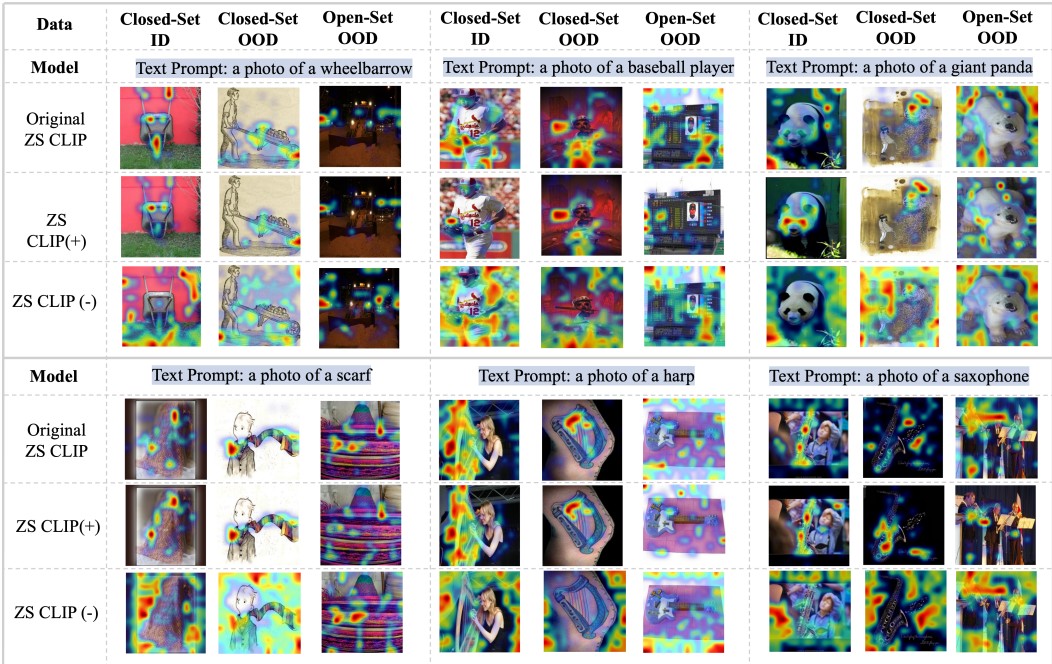

Figure 3: The significant prediction difference between closed-set data and open-set OOD data when vision-language re-alignment is applied to the zero-shot CLIP model (Radford et al., 2021). This difference offers a novel approach to distinguishing between closed-set and open-set classes. Based on the element-wise product between CLIP's image and text features, the masked ZS CLIP(+) model zeroes out the elements of the image feature where the corresponding values in the product are negative. In contrast, the opposite operation is applied in ZS CLIP(-). It is observed that masking the elements where $P_j < 0$ preserves the model's original attention, which motivates us to leverage this consistency between the original and masked domains to improve OOD generalization.

0.2 for Setup-II, and we use the bottleneck adapter with a hidden dimension that is 1/4 of the original feature dimension. This hyperparameter configuration has been demonstrated as the most effective for generic image datasets, such as ImageNet, in the original research (Gao et al., 2023).

For tuning-based OOD detection methods, we adopt the recommended hyperparameter settings for LoCoOp, NegPrompt, and GalLoP. For LoCoOp (Miyai et al., 2024b), following the original paper, we set $\lambda = 0.25$. The hyperparameter $K$ is searched over the range [100, 200] for Setup-I, and [2, 3, 4, 5] for Setup-II, based on validation data. For GalLoP, we follow its publicly available source code and adopt the same hyperparameter settings as reported in Table 3 of (Lafon et al., 2024), including configurations for local prompts, global prompts, tokens per prompt, and other relevant settings. In the NegPrompt method (Li et al., 2024a), we follow its source code and train the model using all the classes from the ID dataset and train a shared positive prompt and two shared negative prompts w.r.t. each training ID class. The hyperparameters $\beta$ and $\gamma$ are set to 0.1 and 0.05, respectively. In the first stage, CoOp is trained for 100 epochs to obtain the positive prompts. In the second stage, the positive prompts are frozen, and our model is trained for 10 epochs to learn the negative prompts. During the testing phase of LoCoOp, NegPrompt, and GalLoP, we use the GL-MCM score (Miyai et al., 2023) to compute OOD detection results.

For experiments on each method, we repeat 3 times with different random splits to eliminate the effects of randomness. The hyperparameters in each method are selected based on the test accuracy on validation sets.

$\Delta$Energy **based on negative OOD labels** The NegLabel (Jiang et al., 2024) and CSP (Chen et al., 2024) methods introduce massive negative labels to boost OOD detection. The extended label space provides a novel perspective to distinguish OOD samples, leveraging extra clues by examining the similarities between images and the extended labels. The CSP method extends the NegLabel by "make up" the OOD label candidates, which are not standard class names but beneficial for the process.

Table 9: Ablations on the hyperparameter $c$. $\Delta$Energy achieves the overall best performance on both AUROC and FPR95 when $c = 2$.

| Data | c=1 | | c=2 | | c=3 | | c=4 | | c=5 | | c=6 | |
|------|-----|-----|-----|-----|-----|-----|-----|-----|-----|-----|-----|-----|
| | FPR | AUROC | FPR | AUROC | FPR | AUROC | FPR | AUROC | FPR | AUROC | FPR | AUROC |
| ID VS Open-Set OOD | 49.60 | 86.46 | 46.40 | 87.10 | 47.23 | 86.85 | 47.93 | 86.44 | 48.13 | 86.06 | 49.97 | 85.69 |
| Closed-set OOD VS Open-Set OOD | 66.54 | 78.49 | 67.16 | 78.68 | 68.30 | 78.43 | 68.40 | 78.07 | 69.04 | 77.79 | 69.27 | 77.50 |

Table 10: Performance of ImageNet-1K-trained model on the test sets with covariate shifts (such as ImageNet_V2, ImageNet_R, ImageNet_A, and ImageNet_S) and concept shifts (such as ImageNet-Superclass).

| Dataset | ImageNet (ID) | ImageNet_V2 | ImageNet_R | ImageNet_A | ImageNet_S | ImageNet-Superclass | Avg OOD Acc |
|---------|---------------|-------------|------------|------------|------------|---------------------|-------------|
| CLIP | 68.80 | 73.97 | 46.09 | 47.77 | 60.90 | 33.18 | 52.38 |
| CoOp | **71.86** | 76.00 | 48.34 | 50.13 | 64.23 | 36.90 | 55.12 |
| CoCoOp | 71.10 | 76.18 | 48.75 | 50.63 | 64.07 | 37.17 | 55.36 |
| CoOp+EBM (Ours) | 71.70 | **77.10** | **49.02** | **51.35** | **64.78** | **38.24** | **56.10** |

Thus, we also extend our $\Delta$Energy method to CSP, leveraging the informative information from extra OOD labels. Let $\mathcal{Y}_{\text{in}}$ denote the ID labels and $\mathcal{Y}_{\text{OOD}}$ denote the OOD labels. Thus, $\Delta$Energy is calculated as follows:

$$\Delta\text{Energy}(\mathbf{x_i}) = \Delta\text{Energy}(\mathbf{x_i}; \mathcal{Y}_{\text{in}}) - \Delta\text{Energy}(\mathbf{x_i}; \mathcal{Y}_{\text{OOD}}) \tag{36}$$

**Ablations on the hyperparameter $c$** We conduct ablation studies on the hyperparameter $c$, focusing on its effect on the discrimination between closed-set data and open-set OOD data using ImageNet-1k. The AUROC and FPR95 performances are reported in Table 9. The results demonstrate that: 1) There is a trade-off between the AUROC and FPR95. 2) $\Delta$Energy achieves the overall best performance on both AUROC and FPR95 when $c = 2$. Therefore, unless otherwise specified, we set $c = 2$ in $\Delta$Energy for all experiments.

**OOD generalization performance under concept shift** We conduct experiments on the concept-shifted ImageNet-Superclass dataset (Xiao et al., 2024; Santurkar et al., 2020), where each image is annotated with its corresponding superclass label. For the superclass labels, we use the open-source annotations provided in `https://github.com/s-kumano/imagenet-superclass/blob/main/superclass_names.txt`. In Table 10, we report the performance of our ImageNet-1K-trained model on test sets annotated with superclasses. From Table 10, our EBM-based method outperforms baseline models under both covariate and concept shifts. These results demonstrate the effectiveness of our proposed approach in learning domain-invariant features, thereby enhancing the model's ability to generalize under various distribution shifts.

**Fine-tuning accuracy of the proposed EBM on standard datasets used in CLIP** We also evaluate the effect of the proposed EBM loss on fine-tuning accuracy across 11 standard datasets used in CLIP. We implement the EBM loss on PromptSRC and train the models using 16-shot settings with a ViT-B/16 backbone. We train 50 epochs for Imagenet and 200 epochs for other datasets with SGD, following the same training setup as in PromptSRC. The EBM hyperparameter was set to $p\% = 0.6$, and we searched over the values of $\{0.1, 0.5, 1.0, 2.0\}$.

The performance results are reported in Table 11, demonstrating that the proposed EBM method further improves test accuracy on PromptSRC. Since the objective of EBM is to minimize $\mathcal{L}_{\Delta E}$, it encourages the model to make equally high-confidence predictions for both the original and partially masked features, thus facilitating the learning of domain-invariant features between the original domain and the masked domain. The improvements observed in Table 11 empirically suggest that regularization on masked features effectively enhances the model's generalization performance.

# H Limitations and Future Work

As demonstrated in Theorems 3.2-3.3 and B.1, our method outperforms the MCM approach by enlarging the difference between the ID and open-set OOD samples. The empirical results in Tables 1–2 and Tables 6–7 further support our theoretical findings. While our method also improves performance on conventional OOD detection benchmarks, the gains are less pronounced compared to the hard OOD scenarios. This may be due to the reduced impact of amplifying the distinction between

Table 11: Performances of fine-tuning accuracy on 11 standard datasets used in CLIP.

| Data | ImageNet | Caltech101 | OxfordPets | Cars | Flowers102 | Food101 | Aircraft | SUN397 | DTD | EuroSAT | UCF101 | Avg |
|------|----------|-----------|-----------|------|-----------|---------|----------|--------|-----|---------|--------|-----|
| CLIP | 66.7 | 92.2 | 88.4 | 65.5 | 70.7 | 84.8 | 24.8 | 62.3 | 44.1 | 48.3 | 64.7 | 64.8 |
| CoOp | 71.7 | 95.6 | 91.9 | 83.1 | 97.1 | 84.2 | 43.4 | 74.7 | 69.9 | 84.9 | 82.2 | 79.9 |
| CoCoOp | 71.0 | 95.2 | 93.3 | 71.6 | 87.8 | 87.2 | 31.2 | 72.2 | 63.0 | 73.3 | 78.1 | 74.9 |
| MaPLe | 72.3 | 96.0 | 92.8 | 83.6 | 97.0 | 85.3 | 48.4 | 75.5 | 71.3 | 92.3 | 85.0 | 81.8 |
| PLOT | 72.6 | 96.0 | 93.6 | 84.6 | 97.6 | 87.1 | 46.7 | 76.0 | 71.4 | 92.0 | 85.3 | 82.1 |
| PromptSRC | 73.2 | 96.1 | 93.7 | 83.8 | 97.6 | 86.5 | 50.8 | 77.2 | 72.7 | 92.4 | 86.5 | 82.8 |
| **Ours** | **73.6** | **96.5** | **94.4** | **85.3** | **98.2** | **87.6** | **51.3** | **77.3** | **73.3** | **93.5** | **86.7** | **83.4** |

Table 12: Comparison of computational efficiency between our method and prior approaches.

| Category | Method | Time | GPU (MB) | Batch Size |
|----------|--------|------|----------|-----------|
| Zero-shot | MCM | 17s | 1706 | 100 |
| | $\Delta$Energy (Ours) | 18s | 1706 | 100 |
| Fine-tuning | CoOp | 18min | 7658 | 32 |
| | LoCoOp | 25min | 8320 | 32 |
| | GalLoP | 140min | 47492 | 32 |
| | EBM (Ours) | 18min | 7756 | 32 |

ID and OOD samples when the inherent difference between ID and open-set OOD samples is already substantial. Future work may explore strategies to further enhance $\Delta$Energy by incorporating CSP's informative negative labels.

# I    Computation Efficiency

The proposed zero-shot OOD detection method does not require fine-tuning of VLM parameters. Instead, it introduces a novel post-hoc scoring function to identify open-set OOD samples. All computations are performed in the latent space during the alignment between vision and language representations, enabling improved OOD detection performances with comparable inference time and computational cost to existing methods such as MCM and raw energy scores. Compared to methods like NegLabel and CSP, $\Delta$Energy does not rely on additional negative labels, making it more computationally efficient. For tuning-based approaches, our proposed EBM method enables joint optimization for both OOD generalization and OOD detection by the novel optimization objective as defined in Equation 8. It re-aligns vision-language representations in the latent space without introducing extra prompts, as required by vanilla CoOp (Zhou et al., 2021), thereby it is more efficient over methods such as LoCoOp (Miyai et al., 2024b) and GalLoP (Lafon et al., 2024). A detailed comparison of computation cost is provided in Table 12. Here, GalLoP is trained using four NVIDIA RTX 4090 GPUs, while all other experiments are conducted on a single NVIDIA RTX 4090 GPU.

