# OpenReview forum: "$\Delta \mathrm{Energy}$: Optimizing Energy Change During Vision-Language Alignment Improves both OOD Detection and OOD Generalization"
_NeurIPS.cc/2025/Conference — NeurIPS 2025 poster_

### Official Review · Reviewer_zrkV · 2025-06-08

**Clarity:** 3
**Significance:** 2
**Originality:** 2
**Rating:** 4
**Confidence:** 4

**Summary:**

This paper introduces **∆Energy**, an OOD detection score for vision-language models based on the energy change after masking top vision-language similarities. The approach combines ∆Energy scoring with an energy bound maximization (EBM) framework during prompt tuning. This jointly improves OOD detection and generalization under covariate shifts. Theoretical analysis and extensive experiments show improved performance over previous methods. The method is efficient and easy to integrate into existing vision-language models.

**Questions:**

In Tables 1 and 2, why do the Energy and MaxLogit performances differ so drastically? In my understanding, their results should be quite similar.

**Ethical Concerns:**

["NO or VERY MINOR ethics concerns only"]

**Final Justification:**

Most of my concerns have been addressed.

**Limitations:**

See the above.

**Paper Formatting Concerns:**

The Related Work section has been moved to the appendix.

**Quality:**

3

**Strengths And Weaknesses:**

**Strengths**

1. The paper is written clearly, and its method is described in a straightforward manner.
2. It provides detailed theoretical derivations supporting the proposed approach.
3. Experiments are thorough and conducted on multiple benchmarks.

**Weaknesses**

1. The motivation is not sufficiently clear: the method is not proposed to address any specific issue with existing methods, nor to fill a well-defined gap.
2. Originality is limited: pruning or zeroing the maximum score has been used in traditional OOD detection (e.g., DICE, ASH) to amplify ID–OOD differences, yet these are neither cited nor compared.
3. I am concerned about the method’s performance: most comparisons are against very basic methods, and the reported gains in Tables 1 and 2 are only marginal(e.g., MSP, MCM). More recent and stronger baselines should be included.
4. I doubt the claim that ΔEnergy outperforms MCM: energy-based scoring already often outperforms MSP (with existing theoretical support), and applying an energy-style normalization to MCM could yield similar improvements. Therefore, it is unclear that the gains stem specifically from the authors’ ΔEnergy formulation.
5. The authors’ method is merely an incremental combination of an ASH-style pruning technique with the Energy approach on top of MCM.

---

> ### Author Rebuttal · Authors · 2025-07-30
>
> >**R1. Reply to ''The motivation is not sufficiently clear'':**
>
> We apologize for any confusion regarding the motivation and the issue we aim to address. We provide clarification as follows:
>
> **Problem Setting of Our Target Issue:**
> In real-world deployment scenarios, models often encounter a mixture of distribution shifts, including both covariate shifts (e.g., changes in style or environment) and semantic shifts (e.g., unseen classes during fine-tuning). To reflect this reality, our work considers a more practical and challenging setting where the test distribution consists of both closed-set OOD data (covariate shifts) and open-set OOD data (semantic shifts).
>
> Under this setting, the model is required to:
> (1) Maintain accurate classification of covariate-shifted samples—a capability referred to as OOD generalization;
> (2) Recognize semantic-shifted OOD samples and abstain from blindly classifying them into any ID class—highlighting the importance of OOD detection.
>
> This setting has attracted increasing interest in recent years [1-4]. Importantly, this problem is especially critical for high-stakes applications (e.g., medical imaging or autonomous driving), where models must simultaneously generalize across domains and detect unseen semantic categories.
>
> **Clarification of the Motivation and Effectiveness of Our Method:**
> We have explained the benefits of $\Delta\mathrm{Energy}$ and EBM for OOD detection and OOD generalization from both theoretical and empirical perspectives. The theoretical motivations are thoroughly discussed in R1 and R2 to Reviewer 3pyx, particularly through the logical flow behind the theorems. Due to space limitations, we kindly refer the reviewer to R2 to Reviewer 3pyx for the in-depth explanation of our theoretical motivations. Empirical evidence supporting the effectiveness of EBM is presented in Tables 3 and 4 of the manuscript, and in Tables 2-1 and 2-2 in response to Reviewer 3pyx, where consistent improvements are observed across both OOD tasks.
>
> [1] Feed two birds with one scone: Exploiting wild data for both out-of-distribution generalization and detection ICML, 2023.
>
> [2] Aha: Human-assisted out-of-distribution generalization and detection. NeurIPS, 2024.
>
> [3] Diversify: A general framework for time series out-of-distribution detection and generalization. TPAMI, 2024
>
> [4] Bridging ood detection and generalization: A graph-theoretic view. NeurIPS, 2024
>
> > **R2. Reply to ''Pruning-based methods are not discussed':**
>
> We apologize for any confusion caused by the lack of discussion on pruning-based OOD detection methods. We will include a corresponding discussion and cite all mentioned papers. Below, we highlight the key differences between our method and related works such as DICE and ASH:
>
> - Different problem setting: Our method connects $\Delta\mathrm{Energy}$ with domain-consistent Hessians and is specifically designed to jointly optimize OOD generalization and OOD detection, backed by theoretical guarantees. In contrast, DICE and ASH primarily target OOD detection by removing noisy weights or activations, without addressing the generalization aspect. Moreover, our method performs V-L re-alignment and explicitly measures energy changes, while prior methods reshape the model outputs through pruning without considering such energy-change criteria.
>
> - Different model assumptions: Our method is tailored for CLIP-based scenarios, where the output logits tend to be uniform-like, posing greater challenges for OOD detection. We particularly focus on hard OOD detection settings, where the semantic gap between ID and OOD samples is small. In contrast, prior works such as DICE and ASH primarily focused on traditional visual models, where the logit corresponding to the ground-truth label is often much higher than the others.
>
> > **R3. Reply to ''Concerns about the method’s performance':**
>
> **Reply to the concern on comparisons:**
>
> We have compared our method with recent CLIP-based state-of-the-art OOD detection methods, including NegLabel (ICLR 2024) and CSP (NeurIPS 2024).  As shown in Tables 1,2, and 7 of our manuscript, our method significantly outperforms both NegLabel and CSP, demonstrating its superiority in distinguishing different semantics in hard OOD detection scenarios. The underlying reasons for the inferior performance of NegLabel and CSP have been discussed in Lines 239-246. In contrast, aligning with the results in Theorem B.1, our method is effective in hard OOD detection scenarios, where the number of classes is high and the similarity gap between closed-set and open-set samples is small.
>
> **Reply to the concern that gains are less significant on some data:**
>
> We attribute the modest gains on PACS and VLCS to two key factors: (1) the inherent discrepancy between ID and easily separable open-set OOD samples is already substantial in these benchmarks; and (2) for hard-to-detect open-set OOD samples, the maximum cosine similarities may not satisfy the condition:
> $s_\hat{y_1}(x_{ID}) > s_\hat{y_1}(x_{OOD})$.
> Thus, the benefit of amplifying ID-OOD separation using $\Delta\mathrm{Energy}$ is diminished. This outcome aligns with our theoretical findings in Theorem B.1, which suggest that $\Delta\mathrm{Energy}$ is more effective in challenging OOD detection scenarios—specifically, when the similarity gap between closed-set and open-set samples is small. Such cases are more prevalent in open-world settings, which we evaluate using ImageNet-based benchmarks. The results, presented in Tables 1, 3, 6, and 7, show substantial performance gains, demonstrating the effectiveness of $\Delta\mathrm{Energy}$ in handling hard OOD scenarios.
>
> > **R4. Reply to ''Doubt on the claim that $\Delta\mathrm{Energy}$ outperforms MCM':**
>
> As shown in the experimental results in Tables 1, 2, and 6 in the manuscript, our method consistently outperforms the energy-based scoring baseline. **The superior performance of $\Delta\mathrm{Energy}$ over the raw energy-based method indicates that the $\Delta\mathrm{Energy}$ formulation itself contributes meaningfully to enhanced OOD detection performance.**
>
> Moreover, in Theorem 3.2, we provide a theoretical justification showing that $\Delta\mathrm{Energy}$ leads to stronger separation between ID and open-set OOD samples than MCM. We emphasize that this theoretical analysis is conducted under CLIP-like model conditions—where the output logits are typically uniform-like (as noted in MCM). This differs significantly from visual models trained with cross-entropy loss, where the logit corresponding to the ground-truth label is often much higher than the others. **As noted in the original MCM paper, for CLIP-like models, applying softmax helps sharpen the uniform-like logits output and increases the separation between ID and OOD samples. By tuning the temperature in the softmax, MCM significantly outperforms the standard energy-based scoring.**
>
> **Taken together, both our theoretical and empirical results support the conclusion that, in the context of CLIP-like models, the $\Delta\mathrm{Energy}$ formulation itself plays a critical role in enhancing OOD detection.**
>
> > **R5. Reply to ''The method is merely an incremental combination of an ASH-style pruning technique with the Energy on top of MCM:**
>
> Our work offers both theoretical advancements and a broader application scope across two OOD tasks within the vision-language model (VLM) context—namely, OOD generalization and OOD detection—rather than simply combining ASH-style pruning with energy-based methods. The key distinctions are as follows:
>
> - Different model assumptions and data settings: We focus on CLIP-based models, where the output logits tend to be uniform-like, making OOD detection more challenging. Moreover, we target hard OOD detection scenarios, where the semantic gap between closed-set and open-set OOD samples is small and the model must robustly handle both covariate and semantic distribution shifts.
>
> - Different methodology: Our method performs vision-language re-alignment and introduces a novel post-hoc OOD score that explicitly measures energy changes. In contrast, ASH modifies model outputs by pruning activations and relies on existing post-hoc OOD scores, such as the energy score or MSP, without incorporating energy-change criteria.
>
> - Unified framework for both OOD generalization and OOD detection: We theoretically prove that the $\Delta\mathrm{Energy}$-based lower-bound maximization (EBM) can theoretically improve both OOD tasks, explicitly increasing $\Delta\mathrm{Energy}$ for closed-set data and learning domain-invariant information between original data and masked data. In contrast, MCM and Energy Score focus solely on OOD detection without addressing OOD generalization, and ASH is an empirical, pruning-based method tailored to visual models trained with cross-entropy loss, lacking theoretical support for application to VLMs.
>
> > **R6. Reply to ''Why do the Energy and MaxLogit performance differ:**
>
> In CLIP-like models, the output logits tend to be uniform-like, meaning that the similarity scores across different classes are relatively close in magnitude. In such cases:
>
> - The Energy score, which is based on the log-sum-exp of all logits, becomes more sensitive to the overall distribution of logits output and less discriminative when the logits are not sharply separated. This can reduce its ability to distinguish between ID and OOD samples.
> - In contrast, MaxLogit focuses solely on the maximum class score, which provides a sharper and more focused signal for confidence—even if all logits are close. This makes it more robust in detecting anomalies when the distribution of scores across classes is flat, as it picks out the highest response directly without aggregating all logits.
>
> Additional empirical evidence of MaxLogit surpassing Energy in CLIP-based cases can be found in Table 1–2 of the CLIPN paper [5].
>
> [5] Clipn for zero-shot ood detection: Teaching clip to say no. ICCV, 2023.

---

> > ### Comment · Reviewer_zrkV · 2025-08-01
> >
> > 1.I remain unconvinced that ∆Energy truly outperforms MCM. For example, directly applying the energy score on CLIP similarity (e.g., using Equation (2)) may already yield similar results, as the theoretical advantage of energy over MSP has been well established in [r1].
> >
> > [r1] Energy-based Out-of-distribution Detection.
> >
> > 2.I still believe the novelty of this paper is quite limited. The method is heavily based on MCM and closely resembles the Energy approach. It remains unclear whether the reported improvements stem from the ∆Energy design itself or simply from the benefits of using energy-based scores in general.

---

> > > ### Author Response · Authors · 2025-08-02
> > >
> > > > **Reply to ''directly applying Energy on CLIP yield similar results'':**
> > >
> > > Directly applying energy score on CLIP similarity results in a 30\% performance degradation compared to ΔEnergy and a 25\% degradation compared to MSP, as shown in the zero-shot results reported in Tables 1 and 2. Note that all results in Tables 1 and 2 are obtained under the CLIP zero-shot setting.
> > >
> > > To clarify, we acknowledge that Energy outperforms MSP when the maximum logit is significantly higher than the others, as illustrated by the examples in lines 5–7 of Table 4-1. We analyze the specific conditions under which this advantage holds in our subsequent response.
> > >
> > > However, as noted in the MCM paper, CLIP’s logits are typically more uniform. In such scenarios, the logit distributions of ID and OOD samples tend to be similar, and the separability of the Energy becomes worse than that of MSP, as shown in lines 2–4 of Table 4-1. In this work, we focus on these harder scenarios and propose ΔEnergy to improve OOD detection. **The best results of ΔEnergy, as shown in Table 1, highlight its effectiveness in hard OOD detection tasks than provious post-hoc methods.**
> > >
> > > > **Reply to ''the theoretical advantage of energy over MSP in [r1]'':**
> > >
> > > First, the advantage of Energy over MSP does not contradict our algorithmic and theoretical contributions. As established in Theorems 3.2 and 3.3, we demonstrate that ΔEnergy outperforms MCM, particularly in CLIP-based hard OOD scenarios.
> > > Second, Energy is not always better than MSP. Based on the analysis in [r1], we analyze the conditions under which the theoretical benefit of Energy over MSP holds. We restate the derivation in [r1] below:
> > >
> > > $$
> > > \max_y p(y \mid \mathbf{x}) = \frac{1}{\sum_i e^{f_i(\mathbf{x}) - f^{\max}(\mathbf{x})}}
> > > $$
> > >
> > > $$
> > > \Rightarrow \log \max_y p(y \mid \mathbf{x})
> > > = \mathbb{E}(\mathbf{x}; f)+ f^{\max}(\mathbf{x})
> > > $$
> > >
> > > This derivation shows that the log of the softmax score is equivalent to a special case of the free energy score, where all the logits are shifted by their maximum logit value. If the Energy score achieves better ID-OOD separability than the maximum logit, then shifting the Energy score by the maximum logit (i.e., forming the log-softmax) actually harms separability, and MSP will underperform Energy.
> > >
> > > However, in more challenging scenarios:
> > >
> > > 1. The sum of exponentials for ID and OOD samples may be similar, making it difficult for Energy to distinguish between them.
> > > 2. In contrast, MSP can still succeed if the maximum logit for the ID sample is slightly higher than that of the OOD sample.
> > >
> > > Thus, Energy score often exhibits worse separability than MSP under these harder conditions. We formally define this case as:
> > >
> > > $$
> > > f\_i(\mathbf{x}\_\text{ID}\^\text{max})>f\_i(\mathbf{x}\_\text{OOD}\^\text{max})
> > > $$
> > >
> > > $$
> > > \sum\_{i=1}^{K} e\^{f_i(\mathbf{x}\_{\text{ID}})} \leq \sum\_{i=1}^{K} e^{f_i(\mathbf{x}\_{\text{OOD}})}
> > > $$
> > >
> > > Then we have:
> > >
> > > $$
> > > \text{Energy}(\mathbf{x}\_{\text{OOD}})\leq \text{Energy}(\mathbf{x}\_{\text{ID}})
> > > $$
> > >
> > > $$
> > > \text{MSP}(\mathbf{x}\_{\text{ID}}) >\text{MSP}(\mathbf{x}\_{\text{OOD}})
> > > $$
> > >
> > > In this case, MSP can still distinguish between ID and OOD, but Energy fails.
> > > We illustrate different cases with specific examples:
> > >
> > > **Table 4-1**
> > > |   | logits| MSP| Energy | ΔEnergy|
> > > | -- | -- | -- | -- | -- |
> > > | ID  | [0.5, 0.4, 0.4, 0.4, 0.4, 0.4] | 0.181 | -2.209 | 0.074  |
> > > | OOD | [0.49, 0.48, 0.48, 0.4, 0.4, 0.4] | 0.175 | -2.434 | 0.070  |
> > > | OOD Detection |  | ✔︎ | ✗  | ✔︎ |
> > > | ID | [2.5, 0.4, 0.1, 0.1, 0.0, 0.0]  | 0.681 | -2.883 | 0.982 |
> > > | OOD | [2.4, 0.1, 0.1, -0.1, 0.0, 0.0] | 0.683 | -2.781 | 0.970 |
> > > | OOD Detection |  | ✗  | ✔︎ | ✔︎ |
> > >
> > > In summary, our method propose a novel OOD score-ΔEnergy-that measures the energy change after re-aligning the CLlP model. We further integrate this formulation to help OOD generalization, with both theoretical and empirical justifications.
> > >
> > > >  **Reply to ''the method closely resembles Energy'':**
> > >
> > > First, we clarify that our method performs vision-language re-alignment and introduces a novel OOD score that explicitly measures energy changes, named ΔEnergy. As shown in Fig 1(C) in the paper, the substantial performance gap between Energy and ΔEnergy reveals that capturing the variation in model behavior provides additional insight for OOD detection that is not available from the raw score alone.
> > >
> > > Second, we emphasize that the ΔEnergy-based lower bound maximization is proposed for both OOD detection and OOD generalization. This formulation is supported by theoretical guarantees, enabling improved robustness under various distribution shifts.
> > >
> > > > **Reply to ''whether the improvement stem from the design itself'':**
> > >
> > > Our analysis of Energy score failure cases, combined with the strong zero-shot performance gains of ΔEnergy under the CLIP setting (up to 30\% in FPR95 as shown in Table 1), provides direct empirical evidence that the improvements stem from the ΔEnergy formulation itself, rather than from the original Energy score.

---

> > > > ### Comment · Reviewer_zrkV · 2025-08-02
> > > >
> > > > I have a few questions and concerns:
> > > >
> > > > 1.Are you sure that the energy score in your paper is computed directly over CLIP similarities rather than classification logits? If it is applied directly to CLIP similarities, then MSP and MCM would essentially be the same, since both apply softmax to the same similarity values.
> > > >
> > > > 2.When 𝑐=1, the ∆Energy score seems very close to the maximum CLIP similarity before softmax. From this perspective, it’s difficult to claim the novelty of the method.
> > > >
> > > > 3.Using the top-k (e.g., top-1, top-2, etc.) CLIP similarities to compute an OOD score seems like a very natural baseline. I believe a comparison with such methods is necessary, as they are closely related to your approach.
> > > >
> > > > 4.I will discuss the paper further with the other reviewers before making a final decision. Thank you for your response.

---

> ### Author Response · Authors · 2025-08-03
>
> ## Reply to Q1:
>
> **The MSP and MCM are different in the paper that arises from the use of different temperature coefficients in the softmax computation.** To clarify, we first define the CLIP similarities and classification logits as follows:
>
> We denote the zero-shot text features as $z_T$ and the zero-shot image features as $z_I$. The CLIP similarity is computed as $\langle z_I, z_T \rangle$, and the CLIP classification logits are computed as $T \cdot \langle z_I, z_T \rangle$, where the temperature coefficient $T$ used by CLIP is 100. In Tables 1 and 2, methods including Energy, MSP, MaxLogit, ReAct, ODIN, and ΔEnergy are computed directly over CLIP’s classification logits, whereas the MCM method is computed based on CLIP similarities (i.e., $T=1$), as recommended in the original MCM paper. Therefore, the discrepancy between MSP and MCM arises from the use of different temperature coefficients in the softmax computation in Tables 1 and 2.
>
> To mitigate the influence of temperature and to better demonstrate the contribution of ΔEnergy under different temperature coefficients, we conduct ablation studies on MSP, Energy, and ΔEnergy. From **Table 4-2**, our method obtains the best results when we use default CLIP classification logits (T=100). Compared to the original Energy scoring method, the performance gains of  ΔEnergy over Energy are around 10\%-35\%, showcasing the effectiveness of ΔEnergy in CLIP-based zero-shot OOD detection.
>
> Table 4-2
>
> |   |   | MSP   | Energy | ΔEnergy   | Performance gain of ΔEnergy over Energy |
> | --- | -- | --- | -- | -- | -- |
> | T = 100 | FPR95 | 53.34 | 76.72  | **46.40** | +30.32   |
> | T = 100 | AUC   | 85.81 | 76.94  | **87.10** | +10.16 |
> | T = 10  | FPR95 | 52.35 | 96.58  | 75.37  | +21.21 |
> | T = 10  | AUC   | 86.24 | 40.14  | 78.13  | +37.99  |
> | T = 1   | FPR95 | 51.72 | 97.14  | 76.67 | +20.47  |
> | T = 1   | AUC   | 85.64 | 39.47  | 77.07 | +37.60  |
> | T=0.1   | FPR95 | 54.51 | 96.09  | 77.92  | +18.17
> | T = 0.1 | AUC   | 85.66 | 41.22  | 76.77 | +35.55  |
>
> ## Reply to Q2:
>
> **When $c=1$, the ΔEnergy score is fundamentally different from the maximum CLIP similarity before softmax (i.e., MaxLogit)—both in formulation, theoretical justification, and empirical performance.** Specifically, the maximum CLIP similarity does not account for the distribution of logits, while ΔEnergy explicitly captures this distribution by computing the energy change before and after re-aligning CLIP. We elaborate on the superiority of our method below:
>
> 1. **Evidence that ΔEnergy outperforms MaxLogit:** As noted in the original MCM paper, for CLIP-like models, applying softmax helps sharpen the uniform-like logits output and increases the separation between ID and OOD samples. As shown in Tables 1 and 2 in the paper, MCM significantly outperforms the MaxLogit method. Since our method theoretically outperforms MCM—as supported by Theorems 3.2 and 3.3—it naturally achieves better results than MaxLogit, which is also empirically demonstrated in Tables 1 and 2.
>
> 2. Furthermore, we emphasize our unified **ΔEnergy-based lower bound maximization framework**, which enhances both OOD detection and OOD generalization. This formulation is supported by theoretical guarantees and leads to improved robustness under various distribution shifts. Our theoretical contribution—which connects OOD generalization and OOD detection through a unified ΔEnergy-based loss—highlights the novelty and the practical significance of ΔEnergy. Detailed theoretical results can be found in Theorem 3.5 and Proposition 3.6.
>
> ## Reply to Q3:
>
> **ΔEnergy is fundamentally different from using the sum of the top-k logits or similarity as OOD score—both in formulation and empirical performance.** The top-k baseline does not consider the overall logit distribution, whereas ΔEnergy captures the distribution by measuring the energy change.
>
> We conducted experiments using the sum of the top-k logits (with k = 1, 2, 3) as OOD scores for ImageNet-based OOD detection. **Note that when we set the temperature $T=1$, the OOD score is computed directly from the CLIP similarities without any scaling.** The results are reported in **Table 4-3**. As shown in the table, ΔEnergy consistently outperforms all top-k logit-based baselines by 30\%-40\%. The substantial performance gap indicates that modeling the variation in energy behaviour offers strong discriminative power for OOD detection—information that is not captured by raw logit or similarity magnitudes alone.
>
> Table 4-3
>
> |     |       | Top-1 (baseline) | Top-2 (baseline) | Top-3 (baseline) | ΔEnergy   |
> | --- | ----- | ----- | --- | ---- | --- |
> | T = 100 | FPR95 | 69.12   | 80.20 | 85.08  | **46.40** |
> | T = 100 | AUC   | 80.28  | 71.96   | 66.53   | **87.10** |
> | T = 10  | FPR95 | 70.34     | 80.22   | 86.95  | 75.37     |
> | T = 10  | AUC   | 80.16  | 71.97 | 66.54 | 78.13  |
> | T = 1   | FPR95 | 70.37 | 80.14  | 85.03    | 76.67 |
> | T = 1   | AUC   | 80.16  | 70.97 | 66.53  | 77.07   |

---

> > ### Comment · Reviewer_zrkV · 2025-08-05
> >
> > Thank you for your response. Most of my concerns have been addressed.
> > I will raise my score to a 4.

---

> > > ### Author Response · Authors · 2025-08-06
> > >
> > > We sincerely thank you for taking the time to review our response. Your insightful suggestions have played a pivotal role in enhancing the overall quality of our paper. We are committed to ensuring that the revised manuscript fully addresses the points you raised. We deeply appreciate your thoughtful feedback once again.

---

### Official Review · Reviewer_C3Gj · 2025-06-28

**Clarity:** 3
**Significance:** 3
**Originality:** 3
**Rating:** 4
**Confidence:** 4

**Summary:**

The paper introduces a novel OOD score, ΔEnergy, which enhances vision-language models' (VLMs) robustness in detecting out-of-distribution (OOD) data. ΔEnergy outperforms existing methods by reducing maximum cosine similarity and improving generalization. It also provides a domain-consistent Hessian for better OOD detection. Experiments show ΔEnergy improves performance by 10%-25% in AUROC. This work looks interesting, with a unique research perspective, good performance, and theoretical support.

**Questions:**

1. Why design deltaE and EBM, and what is the underlying logic behind them.

2. How to determine the parameters for OOD detection in deltaE and whether the parameters are robust.

3. On some datasets, the improvement is acceptable, but on some datasets, the improvement is very small. Please explain the reason.

4. Why EBM？ Is the EBM application scenario established? How to satisfy the conditions of the theorem.

**Ethical Concerns:**

["NO or VERY MINOR ethics concerns only"]

**Final Justification:**

My questions have been partially resolved. However, based on the current status of the submitted paper, I have decided to maintain my current score for the following reasons: 1) The correlation between EBM and the improvement of final classification performance is unclear; 2) Insufficient discussion of parameters, especially in the OOD detection case and the few shot case where N values are taken, it is recommended to use a figure instead of a table; 3) How to verify that the conditions are met without an accurate answer; 4) Some typos, such as Eq (6) at line 147; 5) This article has some contributions but is not well distinguished from MCM in the main text, as the entire energy function design was inspired by MCM; 6) Additionally, it is suggested to theoretically demonstrate that inconsistent changes in \ delta E are indeed valid for OOD.

**Limitations:**

yes

**Paper Formatting Concerns:**

It is ok.

**Quality:**

3

**Strengths And Weaknesses:**

Strengths:

1)  An interesting research starting point.

2) The effectiveness of the method has been supported both theoretically and experimentally.

Weaknesses:

1) This work heavily relies on the MCM method, and the performance improvement is not significant. For example, on PACS and
VLCS, AUC, 93.7% VS 93.5%.

2) The writing is too cumbersome and complex. It is recommended to briefly describe the idea of the theorem in the main text section and provide a more formal description in the appendix. In addition, the energy design in the paper lacks in-depth analysis, including EQ 4. It is not very clear why these designs can help OOD tasks.

3) There is a problem with the application scenario of EBM, which requires obtaining several samples from the source domain. Is there any practical application background? How to satisfy the conditions of its theorem and how to detect its satisfaction.

4) Lack of parameter analysis experiments, especially on how to determine the threshold of delta E in the experiment.

---

> ### Author Rebuttal · Authors · 2025-07-30
>
> >**R1. Reply to ''This work relies on the MCM method, and the improvement is not significant on some data'':**
>
> The proposed $\Delta\mathrm{Energy}$ computes the change in energy score resulting from re-aligning the vision-language modalities by setting the top-2 maximum cosine similarities to zero. We would like to clarify the difference between our $\Delta\mathrm{Energy}$ and the MCM method, and we also provide an explanation for why the performance improvements are less significant on PACS and VLCS. This discussion will be included in a later version of the manuscript to improve clarity and completeness.
>
> **We highlight the key differences between our method and MCM as follows:**
>
> 1. Theoretical support for the OOD detection capability of $\Delta\mathrm{Energy}$ compared to MCM: We have theoretically demonstrated in Theorem 3.2 that the $\Delta\mathrm{Energy}$ score for ID data is consistently larger than that for OOD data, indicating its effectiveness and discriminative power for OOD detection. Compared to MCM and raw energy scores, $\Delta\mathrm{Energy}$ amplifies the gap between ID and OOD data, a property validated by both theoretical analysis (Theorem 3.2) and empirical results (Tables 1, 2, 5 and 7).
> 2. A $\Delta\mathrm{Energy}$-based optimization algorithm (EBM) for both OOD detection and generalization: Further performance gains can be achieved through our $\Delta\mathrm{Energy}$ lower-bound maximization algorithm (EBM). We have both theoretically and empirically shown that EBM not only enhances OOD detection but also secretly optimizes for improved OOD generalization. Theoretical support is provided in Theorem 3.5 and Proposition 3.6. This ''*feed two birds with one scone*” approach is novel compared to MCM, which focuses solely on detection.
>
> **Regarding the observation that improvements are less significant on PACS and VLCS**, we attribute the more modest gains to two key factors: (1) the inherent discrepancy between ID and easily separable open-set OOD samples is already substantial in these benchmarks; and (2) for hard-to-detect open-set OOD samples, the maximum cosine similarities do not satisfy the condition: $s_\hat{y_1}(x_{ID}) > s_\hat{y_1}(x_{OOD})$.
> As a result, the benefit of amplifying this separation using $\Delta\mathrm{Energy}$ is reduced. This outcome is consistent with our theoretical analysis in Theorem B.1 (Lines 662–668), which suggests that $\Delta\mathrm{Energy}$ is more effective in hard OOD detection scenarios—specifically, when the similarity gap between closed-set and open-set samples is small. In contrast, when the distributional shift between closed-set and open-set data is already pronounced, the benefit of enhancing ID–OOD separability through $\Delta\mathrm{Energy}$ diminishes. However, we emphasize that hard OOD scenarios are more prevalent in open-world settings, which we evaluate using ImageNet-based hard OOD detection benchmarks. The results, presented in Tables 1, 3, 6, and 7, show substantial performance gains, demonstrating the effectiveness of our method.
>
> > **R2. Reply to ''Briefly describe the idea of the theorem, and it is not clear why EBM can help OOD tasks'':**
>
> Following your suggestion, we will improve the writing of methodology in the revised paper to improve its readability. Below, we briefly describe the main ideas behind each theorem and explain why Equation 4 is beneficial for OOD tasks.
>
> **Brief description of the idea behind each theorem:**
>
> Theorem 3.2 provides a formal guarantee that $\Delta\mathrm{Energy}$ can provably surpass the widely-used OOD detection method MCM by amplifying the separation between ID and open-set OOD data.
>
> Theorem 3.3 provides a formal guarantee that $\Delta\mathrm{Energy}$  can provably reduce the false positive rate (FPR) in open-set OOD detection compared to MCM.
>
> Theorem 3.4 implies that minimizing the EBM loss can increase the lower bound of $\Delta\mathrm{Energy}$ for closed-set classes, which effectively enhances the separation between ID and open-set OOD data.
>
> Theorem 3.5 demonstrates that minimizing the EBM loss can lead to domain-consistent Hessians of classification loss between original and masked samples, which serves as a strong indicator for OOD generalization.
>
> Proposition 3.6 shows that minimizing EBM provides a bound on the performance gap between ID data and closed-set OOD data, indicating that optimizing $\Delta\mathrm{Energy}$ for ID data implicitly promotes OOD generalization.
>
> **Why Equation 4 can help OOD tasks:**
>
> Equation 4 introduces a $\Delta\mathrm{Energy}$-based bound maximization function (EBM) during the fine-tuning process, which aims at increasing the lower bound of $\Delta\mathrm{Energy}$ score for closed-set classes, as formally established in Theorem 3.4. We explain why Equation 4 is beneficial for OOD tasks from both theoretical and empirical evidence.
>
> Theoretical Perspective: (a) For OOD detection: minimizing Equation 4 effectively increases the $\Delta\mathrm{Energy}$ score for ID data, thereby enhancing the separation between ID and open-set OOD samples. This improved separability facilitates more accurate OOD detection. (b) For OOD generalization: as shown in Theorem 3.5, minimizing Equation 4 leads to domain-consistent Hessians of the classification loss between original and masked samples, which is a strong indicator of OOD generalization.
> **We have also provided detailed and intuitive explanations of the relationship between the EBM and domain-consistent Hessians, as well as the mathematical logic flow, in R2 to Reviewer 3pyx.**
>
> Empirical Evidence: We also empirically demonstrate the impact of the EBM loss on both OOD tasks through an ablation study on the coefficient $\lambda_0$. The results, shown in Table 2-2, indicate that models trained with EBM loss regularization (i.e., $\lambda_0 \neq 0$) achieve consistently better performance in both AUROC and OOD accuracy compared to the baseline without EBM. This confirms the effectiveness of EBM in jointly optimizing for OOD generalization and OOD detection.
>
>
> > **R3. Reply to ''the application scenario of EBM'':**
>
> Our EBM framework follows a common practice in few-shot OOD generalization settings, where only limited source domain samples are available. Few-shot OOD generalization has been extensively studied in prior works. However, these existing methods primarily focus on generalization to closed-set OOD data. In contrast, real-world deployment environments typically involve a mixture of distribution shifts, including both covariate shifts (e.g., changes in style or domain) and semantic shifts (e.g., unseen classes during fine-tuning). To better reflect this reality, our work considers a more practical and challenging setting: the test distribution is composed of both closed-set OOD data (covariate shifts) and open-set OOD data (semantic shifts). This setting has attracted increasing interest in recent years [1-5]. Importantly, this problem is especially critical for high-risk applications (e.g., medical imaging or autonomous driving), where models must simultaneously generalize across domains and detect unseen semantic categories.
>
> [1] Bai, Haoyue, et al. "Feed two birds with one scone: Exploiting wild data for both out-of-distribution generalization and detection." ICML, 2023.
>
> [2] Bai, Haoyue, et al. "Aha: Human-assisted out-of-distribution generalization and detection." NeurIPS, 2024.
>
> [3] Gwon, Kyungpil, et al. "Out-of-distribution (OOD) detection and generalization improved by augmenting adversarial mixup samples." Electronics, 2023.
>
> [4] Zhu, Lin, et al. "CRoFT: robust fine-tuning with concurrent optimization for OOD generalization and open-set OOD detection." ICML, 2024.
>
> [5] Lu, Wang, et al. "Diversify: A general framework for time series out-of-distribution detection and generalization." TPAMI, 2024
>
> > **R4. Reply to ''How to detect the satisfaction of theorem conditions'':**
>
> The key assumption underlying the EBM method is formalized in Equation 7. Since the cosine similarity after re-alignment is defined as $\tilde{s}_{\hat{y}_j}(\mathbf{x_i})= 0$, Equation 7 simplifies to:
>
> $\sum_{i=1}^N e^{s_{\hat{y}_1}(\mathbf{x}_i)/\tau} \geq $
>
> $\sum_{i=1}^N [(e^{\varepsilon_E} - 1) e^{-E_2(\mathbf{x}_i)}+1]$
>
> where $s_{\hat{y}_1}(\mathbf{x_i})$ is the maximum cosine similarity for input $\mathbf{x_i}$, $ \varepsilon_E$ is the upper bound of EBM loss, and $E_2(\mathbf{x_i})$ is the energy score for masked feature.
>
> This inequality implies that, after fine-tuning, the maximum cosine similarity scores for ID samples (i.e., $s_{\hat{y_1}}$) must be sufficiently large. At the same time, it requires the bound $\varepsilon_E$ (in the constraint $\mathcal{L}_{\Delta E} \leq \varepsilon_E$) not to be too loose. These conditions are both reasonable: the first encourages confident classification, while the second ensures effective optimization of the EBM loss. Since all terms in Equation 7 are computable during fine-tuning, the theoretical condition is verifiable in practice.
> Overall, both the practical application scenarios and the verifiable theoretical conditions highlight the broad applicability of our method.
>
> > **R5. Reply to ''Lack of parameter analysis experiments'':**
>
> In our manuscript, we have presented ablation study results for the hyperparameters $c$ and $p$ in Tables 9 and 10, respectively. Additionally, the ablation results for $\lambda_0$ are provided in Table 2-2 of R3 to Reviewer 3pyx. Due to space constraints, we kindly refer the reviewer to our response (R3) to Reviewer 3pyx for details regarding the hyperparameter selection strategy.
>
> > **R6. Reply to Questions:**
>
> A1. The underlying rationale behind $\Delta\mathrm{Energy}$ and EBM is elaborated in R2.
>
> A2. The hyperparameter selection strategy is detailed in R5.
>
> A3. We discussed the reasons why the improvement is small on some datasets in R1.
>
> A4. The practical application scenarios and how to verify theoretical conditions are explained in R3 and R4.

---

> > ### Comment · Reviewer_C3Gj · 2025-08-03
> >
> > My questions have been partially resolved. However, based on the current status of the submitted paper, I have decided to maintain my current score for the following reasons: 1) The correlation between EBM and the improvement of final classification performance is unclear; 2) Insufficient discussion of parameters, especially in the OOD detection case and the few shot case where N values are taken, it is recommended to use a graph instead of a table; 3) How to verify that the conditions are met without an accurate answer; 4) Some typos, such as Eq (6) at line 147; 5) This article has some contributions but is not well distinguished from MCM in the main text, as the entire energy function design was inspired by MCM; 6) Additionally, it is suggested to theoretically demonstrate that inconsistent changes in \ delta E are indeed valid for OOD.

---

> ### Author Response · Authors · 2025-08-03
>
> We thank the reviewer for the valuable feedback. Your suggestion is critical to strengthening our work! Regarding the concerns about the current state of our paper, we provided detailed explanations in our rebuttal. The relevant discussions will be incorporated into the revised version of the paper to improve clarity and completeness. We apologize for any remaining confusion and offer additional clarification below.
>
> **Regarding concern 1:**
>
> As shwon in in response `R2`, we discussed the correlation between EBM and improvements in classification performance from both theoretical and empirical perspectives. To further clarify this relationship, we will incorporate intuitive explanations into the revised version—particularly the connection between EBM and the learning of domain-invariant information, along with illustrative and intuitive explanations of the underlying mathematical reasoning. We thank the reviewer again for raising this important point.
>
> **Regarding concern 2:**
>
> Following your suggestion, in addition to the ablation studies in Tables 9 and 10 of the paper and `Table 2-2` in our response to Reviewer 3pyx, we provide additional ablation experiments on the number of few-shot samples in the ImageNet-based setting. The results are reported in Table 3-1. The consistent improvements observed across both OOD tasks under various few-shot settings demonstrate the robustness of our method. To enhance clarity, we will also provide graphs to visualize the experimental results.
>
> Table 3-1 Ablations on the few shot number.
>
> | DATA | shot = 1  | shot = 1 | shot = 16 | shot = 16 | shot = 32 | shot = 32 |
> | -- | -- | -- | -- | -- | -- | -- |
> | Method  | w / o EBM | w / EBM  | w / o EBM | w / EBM   | w / o EBM | w / EBM   |
> | ID Acc | 79.3  | **79.5** | **82.1**  | 81.5  | **83.0**  | 82.0 |
> | OOD Acc | 60.5 | **61.9** | 61.4 | **63.3**  | 61.5 | **63.5** |
> | FPR95  | 75.0 | **71.6** | 73.2 | **65.9** | 78.1| **68.4** |
> | AUROC  | 73.9 | **77.6** | 72.9 | **81.9** | 67.2 | **79.2** |
>
> **Regarding concern 3:**
>
> We have explicitly explained how to verify the conditions  (Eq 7) of Theorem 3.4 in response `R4`. Since all terms in Equation 7 are computable during fine-tuning, the theoretical condition is practically verifiable.
>
> To further improve clarity, we outline the verification procedures for the theorems related to OOD detection as follows:
>
> - For Theorem 3.2, it is straightforward to verify the condition that the maximum cosine similarity for an ID sample is greater than that of an open-set OOD sample, i.e., $s\_{\hat{y}\_1}(\mathbf{x}\_{\text{ID}}) > s_{\hat{y}\_1}(\mathbf{x}\_{\text{OOD}})$.
> - For Theorem 3.3, under the same data assumptions as in MCM, we only need to check Assumption C.1, which is also easily verifiable in practice, as all required terms in Assumption C.1 are directly computable.
>
> **Regarding concern 4:**
>
> We apologize for the typo. The ∆Energy-based bound maximization is defined in Eq 4, and we will correct this in the revised version of the paper.
>
> **Regarding concern 5:**
>
> **The key differences between our method and MCM as highlighted in `R1` will be incorporated into our paper. Although our method requires computing the maximum similarity, it is fundamentally different from MCM in terms of formulation, theoretical justification, and empirical performance.** The significant performance gap observed in Tables 1 and 2 indicates that modeling the variation in energy behavior provides strong discriminative power for OOD detection—information that cannot be captured by raw logit values or similarity magnitudes alone.
>
> **Regarding concern 6:**
>
> We have provided theoretical insights demonstrating that the proposed ΔEnergy improves both OOD detection and OOD generalization in this paper.
>
> (a) OOD Detection: As shown in **Theorem 3.2**, ΔEnergy provably outperforms MCM by amplifying the separation between ID and open-set OOD data, i.e.,
> $$
> d_{\Delta \mathrm{Energy}} > d_{\text{MCM}}
> $$
>
> $$
> d_{\mathrm{Method}} = S_{\mathrm{Method}}(\mathbf{x_\text{ID}})- S_{\mathrm{Method}}(\mathbf{x_\text{OOD}})
> $$
>
> Furthermore, ΔEnergy-based lower bound maximization (EBM) effectively increases the score for ID data, enhancing the separation between ID and open-set OOD samples. This improved separability facilitates more accurate OOD detection.
>
> (b) For OOD generalization: As shown in **Theorem 3.5**,  ΔEnergy-based lower bound maximization (EBM) leads to domain-consistent Hessians of the classification loss between original and masked samples, which is a strong indicator of OOD generalization. **Proposition 3.6** shows that minimizing EBM provides a bound on the performance gap between ID data and closed-set OOD data, indicating that optimizing ΔEnergy implicitly promotes OOD generalization.
>
> **We sincerely appreciate your constructive feedback, which has greatly helped us strengthen our work. We will ensure that the revised manuscript fully addresses these points and are open to further suggestions.**

---

### Official Review · Reviewer_3pyx · 2025-07-02

**Clarity:** 2
**Significance:** 3
**Originality:** 3
**Rating:** 4
**Confidence:** 3

**Summary:**

This paper proposes a novel method to enhance the robustness of Vision-Language Models (VLMs) on two critical tasks simultaneously: out-of-distribution (OOD) generalization to covariate shifts and OOD detection of semantic shifts. The core contributions of the paper are: 1) A zero-shot OOD detection score, ∆Energy, which is based on the insight that this energy change is substantially larger for in-distribution (ID) data than for OOD data. 2) A unified fine-tuning framework, named EBM (Energy-based Bound Maximization). Minimizing the EBM loss not only increases the lower bound of the ∆Energy score but also encourages the model to learn a domain-consistent Hessian matrix.

Extensive experiments on OOD detection and generalization benchmarks validate the effectiveness of both ∆Energy and EBM.

**Questions:**

1.	Intuitive explanation of the EBM loss: The goal of EBM is to minimize L_∆E = E_2 – E_0. Minimizing their difference intuitively encourages the model to make equally high-confidence predictions for both the original and partially masked features. Could you please explain intuitively how this objective guides the model to learn a "domain-consistent Hessian matrix"?
2.	Hyperparameter Selection: The method introduces new hyperparameters, such as c in ∆Energy and the masking proportion p and loss weight λ_0 in EBM. As shown in Table 10 of the appendix, the choice of p has a noticeable impact. Could you elaborate on the selection strategy for these hyperparameters? Is there a principled way to set them, or do they require extensive validation?

**Ethical Concerns:**

["NO or VERY MINOR ethics concerns only"]

**Final Justification:**

The author has addressed my concerns and I will keep the original score for this paper.

**Limitations:**

yes

**Quality:**

3

**Strengths And Weaknesses:**

Strengths:
1.	Novel Insight: The core idea of the paper—using the "energy change" (∆Energy) after perturbing the vision-language alignment as a basis for OOD detection—is novel.

2.	Unified Framework: Building upon the ∆Energy, the paper integrates optimization of OOD generalization and OOD detection in one framework. Theorem 3.5 links the EBM loss to a domain-consistent Hessian matrix, skillfully provides a strong justification for why this unified approach improves OOD generalization.

Weaknesses:
1.	Clarity of the Methodology: Although the overall idea of the paper is clear, some parts of the methodology (Section 3) are presented in a dense and abstract manner, which can make them difficult to follow.

---

> ### Author Rebuttal · Authors · 2025-07-30
>
> We thank the reviewer for the positive comments and constructive suggestions. Below, we provide detailed responses to all the questions raised:
>
> > **R1. Reply to ''Clarity of the Methodology'':**
>
> We apologize for the confusion and will revise the methodology section in a future version of our manuscript to improve readability. Following reviewers' constructive suggestions, we have provided detailed and intuitive explanations of the EBM loss and outlined the core ideas behind each theorem. A summary of the overall logic flow—illustrating the relationships among $\Delta\mathrm{Energy}$, the EBM loss, and domain-consistent Hessians—is presented in R2, with supporting empirical evidence shown in Table 2-1.
> Below, we briefly summarize the intuition behind each theorem:
>
> Theorem 3.2 provides a formal guarantee that $\Delta\mathrm{Energy}$ can provably surpass MCM by amplifying the separation between ID and open-set OOD data.
>
> Theorem 3.3 provides a formal guarantee that $\Delta\mathrm{Energy}$ can provably reduce the false positive rate (FPR) in open-set OOD detection compared to MCM.
>
> Theorem 3.4 implies that minimizing the proposed EBM loss can increase the lower bound of $\Delta\mathrm{Energy}$ for closed-set classes, which effectively enhances the separation between ID and open-set OOD data.
>
> Theorem 3.5 demonstrates that the EBM loss in Equation 4 can lead to domain-consistent Hessians of classification loss between original and masked samples—an indicator of improved OOD generalization.
>
> Proposition 3.6 shows that minimizing EBM provides a bound on the performance gap between ID data and closed-set OOD data, indicating that optimizing $\Delta\mathrm{Energy}$ for ID data implicitly promotes OOD generalization.
>
> All these intuitive explanations will be incorporated into the revised manuscript to enhance clarity and accessibility.
>
>
> > **R2. Reply to ''Intuitive explanation of the EBM loss to guide the model to learn domain-consistent Hessians'':**
>
> As illustrated in the attention maps in Figure 1(B) and Figure 3 of our manuscript, retaining the top-$p$ proportion of elements in $P_j$ and masking the rest corresponds to masking environment-related information, after which we compute the resulting energy score, denoted as $E_2$. The EBM loss then measures the change between $E_2$ and the original energy $E_0$. The key insight is that masking environment-related information typically induces a small energy change, whereas masking domain-invariant information results in a large energy shift. By optimizing the EBM loss to minimize the energy change before and after masking, the model is in turn encouraged to focus on domain-invariant features, thereby promoting stable classification across domains. Thus, it encourages the Hessians of the classification loss to become domain-invariant as well.
>
> To provide an intuitive understanding of EBM’s impact on learning domain-consistent Hessians and improving OOD generalization, we present the relationship between the EBM loss, Hessian distances between the original and masked domains, and OOD accuracy during the training process. We denote the Hessian of the classification loss with respect to the learnable model parameters for the original data as $H_\mathcal{s}$, and that for the masked data as $H_{\mathcal{s}^\prime}$. Following the prior work [1],  we compute the Hessian distances between the original and masked domains as $\Vert H_\mathcal{s} − H_{\mathcal{s}^\prime}\Vert_F$, where $\Vert \cdot \Vert_F$ denotes the Frobenius norm. Table 2-1 reports these quantities throughout training on the ImageNet-based dataset. We observe that EBM consistently yields smaller Hessian distances and higher OOD accuracies, supporting our claim that EBM promotes domain-invariant optimization dynamics.
>
> [1] Hemati, Sobhan, et al. "Understanding Hessian alignment for domain generalization." ICCV, 2023.
>
>
> Table 2-1. Correlation between Hessian distances ($\Vert H_\mathcal{s} − H_{\mathcal{s}^\prime}\Vert_F$) and OOD accuracies / EBM losses under the EBM regularization during the training on the ImageNet-based dataset used in our main experiment. The OOD accuracies are evaluated on the closed-set subset of the ImageNet-R test data.
>
> | Metric            | Epoch       | 1         | 3        | 5        | 7        | 9        | 11       | 13       | 15       | 17       | 19       |
> | ----------------- | ----------- | --------- | -------- | -------- | -------- | -------- | -------- | -------- | -------- | -------- | -------- |
> | loss_EBM          | w / o EBM   | 0.91      | 0.92     | 0.93     | 0.93     | 0.93     | 0.93     | 0.93     | 0.93     | 0.93     | 0.93     |
> | loss_EBM          | **w / EBM** | **0.91**  | **0.91** | **0.89** | **0.88** | **0.87** | **0.87** | **0.85** | **0.84** | **0.83** | **0.83** |
> | Hessian Distances | w / o EBM   | 25.33     | 4.05     | 5.16     | 5.30     | 5.35     | 5.58     | 5.42     | 5.79     | 5.86     | 6.60     |
> | Hessian Distances | **w / EBM** | **19.98** | **3.24** | **2.97** | **3.17** | **3.56** | **3.47** | **3.81** | **4.23** | **4.26** | **4.35** |
> | OOD Acc           | w / o EBM   | 67.5      | 68.8     | **72.5** | 69.4     | 68.8     | 67.2     | 67.8     | 68.4     | 67.2     | 68.8     |
> | OOD Acc           | **w / EBM** | **70.0**  | **71.6** | 69.1     | **70.9** | **70.6** | **68.8** | **70.9** | **69.4** | **71.9** | **72.8** |
>
> In addition, as shown in Theorem 3.5, we theoretically demonstrate that optimizing the EBM loss leads to domain-consistent Hessians. We briefly introduce the logic behind the theoretical relationships as follows:
> As shown in Equation 31, we establish a connection between the Hessians of the classification loss and the gradient of the energy score with respect to the model parameters.
> By comparing the gradients of the energy scores across the original and masked domains, we derive in Equation 33 that smaller differences in these gradients imply smaller differences in the Hessians of the classification loss across domains.
> Therefore, minimizing the EBM loss not only aligns energy score gradients but also encourages Hessian consistency, thereby improving OOD generalization.
> To make the logic flow clearer, we summarize the logic as follows:
>
> ---
>
> **A. Eq. (31): Classification loss Hessians ←→ Energy score gradients**
> ↓
> **B. Domain-consistent Hessians ←→ Domain-consistent energy score gradients**
> ↓
> **C. Optimizing EBM → Small distance between energy score gradients across original and masked domains**
> ↓
> **D. Optimizing EBM → Small distance between Hessians across original and masked domains (i.e. domain consistent Hessians)**
>
> ---
>
>
> > **R3. Reply to ''Elaborate on the selection strategy for hyperparameters'':**
>
> In our main experiments on open-set discrimination using the large-scale ImageNet-1k dataset, we set $c = 2$ and search for the hyperparameters $p$ and $\lambda_0$ within the ranges $p\\% \in \\{0.4, 0.5, 0.6\\}$ and $\lambda_0 \in \\{0.09, 0.10, 0.11\\}$. **Model selection follows the training-domain validation protocol, where models are trained using few-shot samples under different combinations of $p$ and $\lambda_0$, and the model that achieves the highest accuracy on 4-shot subsets—randomly sampled from the training data—is selected.** In our manuscript, we have presented ablation study results for the hyperparameters $c$ and $p$ in Tables 9 and 10 of our manuscript, respectively. Here, we include the ablation results for $\lambda_0$, provided in Table 2-2. The results show that the EBM method achieves the best overall performance in terms of both AUROC and OOD accuracy when $\lambda_0 = 0.1$. **Notably, models trained with EBM loss regularization (i.e., $\lambda_0 \neq 0$) achieve improved performance in both AUROC and OOD accuracy compared to the baseline model without EBM loss.** This demonstrates the effectiveness of our EBM method in jointly optimizing for both OOD generalization and OOD detection.
>
> Through empirical observations from the ablation study, we found that the masking proportion $p\\%$ exhibits a robust performance sweet spot within the range of 0.5 to 0.6, while the coefficient $\lambda_0$ performs best around 0.1. A masking proportion near 0.6 consistently balances the trade-off between retaining sufficient input information for accurate recognition and introducing meaningful corruption to enhance robustness. Similarly, setting $\lambda_0$ around 0.1 achieves a good balance between maintaining classification performance and enforcing energy consistency between the original and masked data. While these empirical observations provide a principled starting point for hyperparameter selection for other downstream tasks, we acknowledge that the optimal choices may vary slightly across different domains. Our ablation studies offer practical guidance for setting the search space of $p$ and $\lambda_0$ on other datasets.
>
> Table 2-2. Ablations on the hyperparameter $\lambda_0$. In the table, we set $c=2$ and $p\\%=0.6$ according to the optimal hyperparameter observed in Table 9 and Table 10. The EBM method achieves the overall best performance on both AUROC and OOD accuracy when $\lambda_0=0.1$.
>
> | $\lambda_0$ | ID ACC    | OOD ACC   | AUROC     | FPR95     |
> | ----------- | --------- | --------- | --------- | --------- |
> | 0.00        | **82.11** | 61.36     | 75.64     | 73.20     |
> | 0.05        | 81.78     | 61.56     | 77.94     | 71.58     |
> | 0.10        | 81.62     | **63.28** | **80.19** | **65.75** |
> | 0.20        | 81.54     | 62.85     | 77.96     | 69.95     |
> | 0.50        | 81.74     | 61.80     | 78.41     | 69.75     |
> | 1.00        | 81.10     | 61.67     | 79.39     | 68.20     |
> | 2.00        | 81.32     | 62.01     | 79.26     | 71.45     |

---

### Official Review · Reviewer_7zhN · 2025-07-03

**Clarity:** 3
**Significance:** 2
**Originality:** 2
**Rating:** 4
**Confidence:** 4

**Summary:**

Motivated by heuristic observations regarding the energy change of cosine similarity in closed-set and open-set data, this paper introduces a novel out-of-distribution (OOD) score, $\Delta Energy$, which simultaneously improves both OOD detection and OOD generalization performance of vision-language models during prompt fine-tuning. Theoretical analysis is provided to demonstrate that the proposed method guarantees better theoretical performance in OOD detection and generalization compared to the baselines. Evaluation on several OOD detection and generalization benchmarks further demonstrates the effectiveness of the proposed method for classification tasks.

**Questions:**

1. In the proposed method, the objective is designed to raise the lower bound of $\Delta Energy$ to improve OOD detection. However, the effect of increasing this lower bound on classification performance in closed-set data is unclear. I am concerned that this may negatively impact classification performance on closed-set data.

2. At the evaluation stage, the performance under another common out-of-distribution scenario, i.e., concept shift, is not assessed. It is suggested to include this evaluation to determine whether the proposed method can be applied to a broader range of out-of-distribution scenarios.

**Ethical Concerns:**

["NO or VERY MINOR ethics concerns only"]

**Final Justification:**

The authors' clarifications and additional experimental results has effectively addressed most of my concerns. After carefully considering the authors' responses and other reviews' comments, I have decided to raise my original rating to 4.

**Limitations:**

**Limitation:**

1. In this paper, the proposed method is specifically designed, discussed, and evaluated for distribution shift caused by covariate shift (i.e., $P_{train}(X) \ne P_{test}(X)$ and $P_{train}(Y|X) = P_{test}(Y|X)$). However, its performance under another common out-of-distribution scenario, namely concept shift (where $P_{train}(X) = P_{test}(X)$ and $P_{train}(Y|X) \ne P_{test}(Y|X)$), remains unclear, which significantly limits its applicability.

**Paper Formatting Concerns:**

No.

**Quality:**

3

**Strengths And Weaknesses:**

**Strengths:**

1. This paper is well-written and easy to follow. The heuristic observations behind the introduced out-of-distribution score are interesting and provide a straightforward approach to improving both out-of-distribution detection and generalization performance for vision-language models during prompt tuning.

2. A detailed analysis is provided to demonstrate that the proposed method can improve both out-of-distribution detection and generalization performance from a theoretical perspective.

3. The empirical results on commonly used OOD datasets demonstrate the effectiveness of the proposed method in improving both OOD detection and generalization performance compared to existing methods.

**Weaknesses:**

1. In this paper, the proposed method is specifically designed, discussed, and evaluated for distribution shift caused by covariate shift (i.e., $P_{train}(X) \ne P_{test}(X)$ and $P_{train}(Y|X) = P_{test}(Y|X)$). However, its performance under another common out-of-distribution scenario, namely concept shift (where $P_{train}(X) = P_{test}(X)$ and $P_{train}(Y|X) \ne P_{test}(Y|X)$), remains unclear, which significantly limits its applicability.

2. In the proposed method, the objective is designed to raise the lower bound of $\Delta Energy$ to improve OOD detection. However, the effect of increasing this lower bound on classification performance in closed-set data is unclear. I am concerned that this may negatively impact classification performance on closed-set data. It is recommended to verify this by evaluating the fine-tuning accuracy of the proposed method on standard datasets used in CLIP [1].

[1] Radford, Alec, et al. "Learning transferable visual models from natural language supervision." International conference on machine learning. PmLR, 2021.

3. At the evaluation stage, the performance under another common out-of-distribution scenario, i.e., concept shift, is not assessed. It is suggested to include this evaluation to determine whether the proposed method can be applied to a broader range of out-of-distribution scenarios.

---

> ### Author Rebuttal · Authors · 2025-07-30
>
> We thank the reviewer for the constructive comments. Below, we provide detailed responses to all the questions raised:
>
> > **R1. Reply to ''the performance under concept shift remains unclear'':**
>
> In our method, we primarily focus on enhancing the robustness of vision-language models under covariate shifts, while simultaneously enabling the detection of open-set semantic shifts. Specifically, **we keep the class names consistent with those used during training in the test phase—a common and practical setting in real-world applications**—and meanwhile aim to prevent the model from incorrectly assigning open-set semantic-shifted OOD samples to any ID class.
>
> Following your valuable suggestion, we conduct experiments on the concept-shifted ImageNet-Superclass dataset [1], where each image is annotated with its corresponding superclass label. For the superclass labels, we use the open-source annotations provided in [1]. In Table 1-1, we demonstrate examples of ImageNet-1K training data alongside concept-shifted test samples. In Table 1-2, we report the performance of our ImageNet-1K-trained model on test sets annotated with superclasses. From Table 1-2, our EBM-based method outperforms baseline models under both covariate and concept shifts. These results demonstrate the effectiveness of our proposed approach in learning domain-invariant features, thereby enhancing the model's ability to generalize under various distribution shifts. All experimental results and related work [2, 3] on concept shifts will be incorporated into the later version of our manuscript to improve its overall quality.
>
> [1] Soichiro Kumano, et al. Dataset Source: github.com/s-kumano/imagenet-superclass/blob/main/superclass_names.txt
>
> [2] Xiao, Zehao, et al. "Any-shift prompting for generalization over distributions." CVPR, 2024.
>
> [3] Santurkar, Shibani, et al. "Breeds: Benchmarks for subpopulation shift." *arXiv preprint arXiv:2008.04859* (2020).
>
> Table 1-1.  Examples of the ImageNet-1K training data and the corresponding concept-shifted test data.
>
> | DATA               | Imagenet-1k    | Concept Shift           |
> | :----------------- | -------------- | ----------------------- |
> | Total Class Number | 1000           | 558                     |
> | **Folder**         | **Class Name** | **Class Name**          |
> | n01440764          | tench          | cyprinid, cyprinid fish |
> | n01443537          | goldfish       | cyprinid, cyprinid fish |
> | n01530575          | brambling      | finch                   |
> | n01531178          | goldfinch      | finch                   |
> | n01532829          | house finch    | finch                   |
> | n01534433          | junco          | finch                   |
>
>
> Table 1-2. Performance of ImageNet-1K-trained model on the test sets with covariate shifts and concept shifts.
>
> | Dataset             | ImageNet (ID) | ImageNet_V2  (covariate shift) | ImageNet_R (covariate shift) | ImageNet_A (covariate shift) | ImageNet_S (covariate shift) | ImageNet-Superclass (concept shift) | Avg OOD Acc |
> | ------------------- | ------------- | ------------------------------ | ---------------------------- | ---------------------------- | ---------------------------- | ----------------------------------- | ----------- |
> | CLIP                | 68.80         | 73.97                          | 46.09                        | 47.77                        | 60.90                        | 33.18                               | 52.38       |
> | CoOp                | **71.86**     | 76.00                          | 48.34                        | 50.13                        | 64.23                        | 36.90                               | 55.12       |
> | CoCoOp              | 71.10         | 76.18                          | 48.75                        | 50.63                        | 64.07                        | 37.17                               | 55.36       |
> | **CoOp+EBM (Ours)** | 71.70         | **77.10**                      | **49.02**                    | **51.35**                    | **64.78**                    | **38.24**                           | **56.10**   |
>
>
>
> > **R2. Reply to '' I am concerned that EBM may negatively impact classification performance on closed-set data.  It is recommended to verify the fine-tuning accuracy of the proposed method on standard datasets used in CLIP'':**
>
> As defined in Equation 4, the proposed objective function—namely, the $\Delta\mathrm{Energy}$-based bound maximization (EBM)—aims to increase the lower bound of the $\Delta\mathrm{Energy}$ for closed-set classes, as demonstrated in Theorem 3.4. We have theoretically shown its impact on closed-set OOD data, particularly on data exhibiting covariate shifts, in Theorem 3.5. The results in Theorem 3.5 indicate that optimizing Equation 4 leads to domain-consistent Hessians of the classification loss, which implies improved model generalization on closed-set data [4, 5].
> Empirically, this theoretical advantage is supported by the OOD accuracy gains observed in Tables 3 and 4 of our manuscript, as well as by the ablation study on EBM presented in Table 2-2 of our response to Reviewer 3pyx, further highlighting the effectiveness of EBM in enhancing OOD generalization.
>
> Following your insightful suggestion, we evaluate the effect of the proposed EBM loss on fine-tuning accuracy across 11 standard datasets used in CLIP. We implement the EBM loss on PromptSRC [6] and train the models using 16-shot settings with a ViT-B/16 backbone. We train 50 epochs for Imagenet and 200 epochs for other datasets with SGD, following the same training setup as in [6]. The EBM hyperparameter was set to $p\\% = 0.6$, and we searched $\lambda_0$ over the values of $\\{0.1,0.5,1.0,2.0\\}$.
>
> The performance results are reported in Table 1-3, demonstrating that the proposed EBM method further improves test accuracy on PromptSRC. Since the objective of EBM is to minimize $\mathcal{L}_{\Delta E}=E_2 - E_0$, it encourages the model to make equally high-confidence predictions for both the original and partially masked features, thus facilitating the learning of domain-invariant features between the original domain and the masked domain. The improvements observed in Table 1-3 empirically suggest that regularization on masked features effectively enhances the model’s generalization performance.
>
> [4] Rame, Alexandre, et al. "Fishr: Invariant gradient variances for out-of-distribution generalization." ICML, 2022.
>
> [5] Hemati, Sobhan, et al. "Understanding hessian alignment for domain generalization." ICCV, 2023.
>
> [6] Khattak, Muhammad Uzair, et al. "Self-regulating prompts: Foundational model adaptation without forgetting." ICCV, 2023.
>
>
> Table 1-3. Performances of fine-tuning accuracy on 11 standard datasets used in CLIP.
>
> | Data      | ImageNet | Caltech101 | OxfordPets | Cars     | Flowers102 | Food101  | Aircraft | SUN397   | DTD      | EuroSAT  | UCF101   | Avg      |
> | --------- | -------- | ---------- | ---------- | -------- | ---------- | -------- | -------- | -------- | -------- | -------- | -------- | -------- |
> | CLIP      | 66.7     | 92.2       | 88.4       | 65.5     | 70.7       | 84.8     | 24.8     | 62.3     | 44.1     | 48.3     | 64.7     | 64.8     |
> | CoOp      | 71.7     | 95.6       | 91.9       | 83.1     | 97.1       | 84.2     | 43.4     | 74.7     | 69.9     | 84.9     | 82.2     | 79.9     |
> | CoCoOp    | 71.0     | 95.2       | 93.3       | 71.6     | 87.8       | 87.2     | 31.2     | 72.2     | 63.0     | 73.3     | 78.1     | 74.9     |
> | MaPLe     | 72.3     | 96.0       | 92.8       | 83.6     | 97.0       | 85.3     | 48.4     | 75.5     | 71.3     | 92.3     | 85       | 81.8     |
> | PLOT      | 72.6     | 96.0       | 93.6       | 84.6     | 97.6       | 87.1     | 46.7     | 76.0     | 71.4     | 92.0     | 85.3     | 82.1     |
> | PromptSRC | 73.2     | 96.1       | 93.7       | 83.8     | 97.6       | 86.5     | 50.8     | 77.2     | 72.7     | 92.4     | 86.5     | 82.8     |
> | **Ours**     | **73.6** | **96.5**   | **94.4**   | **85.3** | **98.2**   | **87.6** | **51.3** | **77.3** | **73.3** | **93.5** | **86.7** | **83.4** |
>
>
> >  **R3. Reply to ''Whether the proposed method can be applied to a broader range of OOD scenarios'':**
>
> We thank the reviewer for the insightful comment. Following your insightful suggestion, we have conducted additional experiments on concept shifts and evaluated model's fine-tuning accuracy on 11 standard datasets used in CLIP. The improvements observed in Tables 1-2 and Table 1-3 demonstrate that the proposed EBM loss, which is designed to facilitate the learning of domain-invariant features between the original and masked domains, can effectively enhance the model’s generalization performance. All experimental results reported in Tables 1-2 and 1-3, along with the related work on concept shifts [2, 3], will be incorporated into a later version of our manuscript to improve its overall quality.

---

> > ### Comment · Reviewer_7zhN · 2025-08-06
> > **Response to the Rebuttal**
> >
> > I sincerely appreciate the authors' clarifications and additional experimental results. Their rebuttal has effectively addressed most of my concerns. Accordingly, I would like to increase my original rating to 4.

---

> > > ### Author Response · Authors · 2025-08-08
> > >
> > > We sincerely appreciate your thoughtful suggestions regarding additional experiments on concept shift and standard CLIP datasets. We’re glad to hear that our responses have addressed your concerns. Your insights have played a pivotal role in improving the overall quality of our paper. Thank you once again for your time and for raising the score.

---

### Decision · Program_Chairs · 2025-09-17

**Decision:**

Accept (poster)

**Comment:**

This paper introduces ∆Energy out-of-distribution (OOD) score for VLMs. ∆Energy is motivated by heuristic observations that the energy change is larger for in-distribution (ID) data than for OOD data. This paper also proposes a fine-tuning framework, named Energy-based Bound Maximization (EBM). Experiments are shown in OOD detection and generalization benchmarks.

This paper has four borderline recommendations, but slightly learning towards positive. The reviewers found that the proposed method is well-motivated and sound (3pyx, 7zhN, C3Gj), and theoretically supported (7zhN, C3Gj).

Meanwhile, there were concerns related to the clarity and the presentation of the paper (3pyx, C3Gj), scope of evaluation (7zhN, C3Gj), and hyperparameter sensitivity (3pyx, C3Gj).

During the discussion period, the rebuttal document provides new experiments on concept shift to handle the evaluation scope issue, and a hyperparameter ablation study.

In summary, this paper has meaningful novelty, theoretical grounding, and empirical results. Also, the most critical concerns were addressed during rebuttal. Overall, I lean towards accept. While clarity and parameter analysis could be further improved, the contributions are meaningful for the vision-language and OOD communities.